| 1        | Advantages of using multiple Doppler radars with different wavelengths for                                        |
|----------|-------------------------------------------------------------------------------------------------------------------|
| 2        | three dimensional wind retrieval                                                                                  |
| 3        |                                                                                                                   |
| 4        |                                                                                                                   |
| 5        |                                                                                                                   |
| 6        |                                                                                                                   |
| 7        |                                                                                                                   |
| 8        | Chia-Lun Tsai <sup>1</sup> , Kwonil Kim <sup>2</sup> , Yu-Chieng Liou <sup>3</sup> , and GyuWon Lee* <sup>2</sup> |
| 9        |                                                                                                                   |
| 10       |                                                                                                                   |
| 11       | <sup>1</sup> Department of Atmospheric and Geological Sciences, Chinese Culture University, Taipei,               |
| 12       | Taiwan                                                                                                            |
| 13       |                                                                                                                   |
| 14       | <sup>2</sup> BK21 Weather Extremes Education & Research Team, Department of Atmospheric Sciences                  |
| 15       | Center for Atmospheric REmote sensing (CARE), Kyungpook National University, Daegu,                               |
| 16       | South Korea                                                                                                       |
| 17       |                                                                                                                   |
| 18       | <sup>3</sup> Department of Atmospheric Sciences, National Central University, Jhongli, Taiwan                     |
| 19       |                                                                                                                   |
| 20       |                                                                                                                   |
| 21       |                                                                                                                   |
| 22<br>23 |                                                                                                                   |
| 23<br>24 |                                                                                                                   |
| 24<br>25 |                                                                                                                   |
| 25<br>26 |                                                                                                                   |
| 20<br>27 | Revised                                                                                                           |
| 28       | Atmospheric Measurement Techniques                                                                                |
| 29       | 11 October 2025                                                                                                   |
|          | = <del></del>                                                                                                     |

<sup>\*</sup> Corresponding author: Prof. GyuWon Lee, E-mail: gyuwon@knu.ac.kr

#### Abstract

The Wind Synthesis System using Doppler Measurements (WISSDOM) is a practical scheme employed to derive high-resolution three-dimensional (3D) winds using any number of radars. This study evaluated the advantages of using multiple radars with different wavelengths in WISSDOM for the analysis of bow-shaped convection in a severe squall line recorded on 2 August 2020. A total of 11 radars were in operation in the areas surrounding Seoul metropolitan, South Korea: four S-band, two C-band, and five X-band radars. The advantages of using these radars were assessed using six different synthesis scenarios: 1) four S-band (scenario S), 2) two C-band (scenario C), 3) five X-band (scenario X), 4) a combination of four S- and two C-band (scenario SC), 5) four S- and five X-band (scenario SX), and 6) four S-, two C-, and five X-band radars (scenario SCX). The results revealed that scenario S offered good coverage in the synthesis domain, but relatively fewer observations were produced near the surface. In contrast, scenarios C and X provided sufficient data at lower levels but less coverage in the areas far from the radars. The scenarios SC and SX captured the return flow at low levels similar to typical squall line structures. Overall, the scenario SCX led to the optimal synthesis when compared with the observations. The mean bias (MB) of the U- and V-winds between the sounding observations and scenario SCX was 0.7 and 0.5 m s<sup>-1</sup>, respectively, while the root mean square difference (RMSD) of the U- and V-winds were around 1.7 m s<sup>-1</sup>. In addition, when comparing the retrieved WISSDOM winds with three radar wind profiler observations, the average MB (RMSD) for the U-, V-, and W-winds was 1.4, 2.0, and 1.0 m s<sup>-1</sup> (3.1, 3.9, and 1.5 m s<sup>-1</sup>), respectively. The significant differences between scenarios S and SCX can be attributed to additional low-level observations in SCX, which allowed for the capture of stronger updrafts in the convection areas of the squall line. Overall, these results highlight the advantages of using radars with multiple wavelengths in WISSDOM, especially C- and X-band radars.

#### 1. Introduction

Doppler radars are important sources of information for weather analysis because of their relatively wide coverage and high spatiotemporal resolution. In particular, meteorological radars are widely used to measure radar reflectivity and radial velocity for determining precipitation structures and kinematic information of the weather systems. Armijo (1969) developed a theory for determining the winds and precipitation vortices using Doppler radar. However, a single Doppler radar can only provide the radial velocity, making it difficult to completely resolve the horizontal and vertical winds in precipitation systems. Miller and Strauch (1974) retrieved three-dimensional (3D) winds in precipitation systems using dual Doppler radars. Nevertheless, due to the insufficient availability of radars, a single Doppler radar was still adopted to investigate the kinematic structure of precipitation systems from the 1980s to the 2000s. In this approach, the mean winds used to analyze the wind patterns of weather systems are usually derived from a single Doppler radar using velocity azimuthal display (VAD; Browing and Wexler, 1968) and velocity track display (VTD; Lee et al., 1994), a technique from which many other methods have been derived, including ground-based VTD (GBVTD; Lee et al., 1999), extended GBVTD (EGBVTD; Liou et al., 2006), and generalized VTD (GVTD; Jou et al., 2008).

Since the 2000s, dual-Doppler synthesis has emerged as a more accurate means to derive complete wind fields if two or more radars are available. The most widely used dual-Doppler retrieval technique is Cartesian Space Editing and Display of Radar Fields under Interactive Control (CEDRIC; Mohr and Miller, 1983), which simultaneously solves equations using observations of the radial velocity from two radars to derive horizontal winds (i.e., U- and V-winds). Vertical winds are then estimated by integrating a continuity equation for the derived horizontal winds, ultimately constructing complete 3D winds. However, CEDRIC has a limitation in that the horizontal winds cannot be completely derived along the radar baseline. To address this limitation and obtain complete wind information, there has been a shift towards using multiple Doppler radars if available. In particular, starting in the 2010s, mathematically

variational approach techniques have been utilized to retrieve winds. For example, Collis et al. (2013) and Varble et al. (2014) used variational techniques to retrieve winds via scanning Doppler radar. In addition, the 3D variational techniques (3DVAR) for radar wind retrieval have been developed by Shapiro and Potvin and are now available on the Python platform PyDDA (Jackson et al. 2020). However, the terrain effects was not significantly considered in their schemes. Liou and Chang (2009) first proposed the Wind Synthesis System using Doppler Measurements (WISSDOM), while Bell et al. (2012) introduced Spline Analysis at Mesoscale Utilizing Radar and Aircraft Instrumentation (SAMURAI) and Cha et al. (2021) applied this scheme in the analysis of a hurricane. Cha and Bell (2023) subsequently upgraded SAMURAI by implementing immersed boundary method (IBM; Tseng and Ferziger, 2003) to more effectively retrieve wind over complex terrain. In addition, Chong and Bousquet (2001) developed the Multiple-Doppler Synthesis and Continuity Adjustment Technique (MUSCAT). These variational techniques considered terrain effects by employing the immersed boundary method (IBM; Tseng and Ferziger, 2003). One of the advantages of this approach is that winds can be recovered along the radar baseline, and high-quality winds can also be derived over complex terrain (Liou et al., 2012, 2013, 2014, 2016; Lee et al., 2018).

Although the quality of the winds derived from WISSDOM is high, sufficient radar observations are required to expand the study domain for specific mesoscale convection systems such as typhoons, long squall lines, winter storms, and windstorms. (Tsai et al., 2022, 2023; Swastiko et al., 2024). Radar observations are generally affected by the terrain because mountains can block the radar beams. Ideally, the use of more radars can minimize this issue because more complete coverage is possible, eliminating blind spots. For example, Tsai et al. (2018) used six radars in WISSDOM—three S-band (wavelength of ~10 cm) and three C-band (wavelength of ~5 cm) radars—to document the mechanisms associated with winter precipitation over the Pyeongchang mountains in South Korea, with detailed precipitation structures and 3D winds successfully retrieved. Although S-band radar usually covers a wide area, radar data may be

missing at lower levels far from the radar site. At the same time, the radar gate volumes become larger if the gate locations are too far from the radar site, leading to ambiguous radar observations, which is why the combination of radars is important. In addition, compared to short-wavelength radars such as C-band or X-band (wavelength of ~3 cm) radars, the coarser spatial resolution of long-wavelength radar observations is less valuable when attempting to resolve precise winds using the fine grid spacing of WISSDOM (Tsai et al., 2022).

Increasing the number of radars or lidars can reduce most concerns about data coverage in wind retrieval algorithms (Choukulkar et al., 2017; Tsai et al., 2023), especially in mountainous areas (Hill et al., 2010). The high construction cost of S-band radar makes it difficult to install them in large numbers and limits their rapid deployment. In addition, the S-band radar is installed on the top of high mountains to secure good coverage, resulting more prone to ground clutter contamination. In contrast, C- and X-band radars are less expensive and more mobile and mor sensitive to smaller precipitation particles. Radars with shorter wavelengths are ideal for gap-filling applications and provide more information in light rain events. Even in areas of light rain, the use of these radars can maintain high-quality wind retrieval. Furthermore, the attenuation issues inherent to short wavelength radars do not affect radial (Doppler) velocity measurements.

Recent advances have underscored the value of enhancing conventional radar networks with additional gap-filling short wavelength radars. For example, Beck and Bousquet (2013) demonstrated that supplementing a national network with X-band radars can substantially improve low-level wind retrieval and extend coverage in complex terrain. Junyen et al. (2010), Bharadwaj et al. (2010) have proposed the application of X-band radar networks deployed by the Center for Collaborative Adaptive Sensing of the Atmosphere (CASA). However, their study primarily focused on the expansion of observational coverage in complex terrain and the qualitative aspects of wind field improvement. A systematic and quantitative analysis using independent observational data is needed to assess the impact of using additional short-wavelength radars. Additionally, there remains a gap in understanding whether the dynamics and

vertical structure of a specific precipitation system can be effectively captured.

In cases where the WISSDOM is specifically used, Liou and Chang (2009) were the first to combine two S-band radars and one X-band radar for WISSDOM, but most research has employed three S-band radar observations in Taiwan (Liou et al., 2012, 2014, 2019, 2024). Liou et al. (2013) also adopted one S-band and one C-band radar in WISSDOM to investigate Typhoon Morakot (2009), while Lee et al. (2018) documented the orographic enhancement of precipitation on Jeju Island, South Korea, using two S-band radar observations. Tsai et al. (2018) used three S-band and three C-band radars to examine the mechanisms of winter precipitation along the northeastern coast of South Korea. Three radars with different wavelengths were adopted by Liou et al. (2016), who used two S-band, one C-band, and one X-band radars in WISSDOM and reported good retrieval results. However, their study remains the only one to date that has combined three different radar wavelengths for WISSDOM, thus the specific advantages of doing so remain unclear.

Recently, Liou et al. (2019) and Liou and Teng (2023) derived thermodynamic fields using retrieved winds from WISSDOM, with the accuracy of the results linked to the data quality of the radar observations. As radar networks continue to expand, high spatiotemporal resolution 3D winds and thermodynamic fields will become increasingly accessible. However, understanding of the benefits of the use of Doppler radars with different wavelengths for the analysis of storm dynamics and phenomena and the mechanisms remains limited. To address this gap, this study conducts a quantitative and systematic assessment of the advantages of using multiple wavelength radars, including their ability to provide more coverage (especially at lower levels) and produce observations with a high spatial resolution. A squall line case was chosen for this evaluation because the presence of significant precipitation and strong winds can be used to examine potential errors in retrieved winds (Tsai et al. 2023). It also allows us to evaluate the uncertainty, and accuracy of wind retrieval using independent wind observations. Additionally, it examines the role of additional short-wavelength radars in capturing the dynamics and vertical

structure of precipitation systems. To achieve this, this study retrieves winds with different synthesis scenarios with a total of 11 radars, including four S-band, two C-band, and five X-band radars.

# 2. Data and methodology

### 2.1 Synthesis domain and observational data

This study focused on the region around Seoul metropolitan areas, South Korea, Seoul, which has the highest population density in the country and a dense radar network. Eleven radars were in operation within the WISSDOM analysis domain, with their locations presented in Fig. 1. The four S-band long-wavelength radars are labeled SBRI, SGDK, SKWK, and SKSN in Fig. 1a, while the automatic weather stations (AWSs), sounding, radar wind profilers (RWPs), and Cand X-band radar sites in the WISSDOM domain are presented in Fig. 1b. The two C-band radars are labeled CIIA and CSAN and the five X-bands radars are labeled XYOU, XKOU, XSRI, XMIL, and XDJK. The temporal resolution for each radar volume scan was 10 min except for CIIA (XDJK, XMIL, and XSRI), which was around 6-7 min (~15 min). A complete volume scan can be synchronized every 30 min for the selected radars. In a complete volume scan of each radar, the plan position indicator (PPI) elevation angles were concentrated between -0.4° and 20° (45° for CIIA), with details of the elevation angles presented in Table 1. Even though they were operated from different governments or university departments, the radars used in this study were mostly synchronized using similar scanning strategies. The gate spacing was between 60 and 250 m, with a maximum range of 40-250 km depending on the wavelength of the radar. The specifications for the radars are summarized in Table 1.

Figure 1. Spatial distribution of instruments used in the present study. A small box in (a) indicates the WISSDOM synthesis domain corresponding to (b). The black triangles denote the radars, the red solid circles indicate the automatic weather stations (AWS) and the black squares represent the sounding (47199) and radar wind profiler sites (RWP1–3). The topographic features and elevation are depicted in accordance with the color scale on the right.

Table 1. Specifications for the radars used in the present study

|      |                |               |                        |                  |                      | 1                                                   |                            |                      |                                     |                                                      |
|------|----------------|---------------|------------------------|------------------|----------------------|-----------------------------------------------------|----------------------------|----------------------|-------------------------------------|------------------------------------------------------|
|      | Longitude (°E) | Latitude (°N) | Radar<br>Height<br>(m) | Wave length (cm) | Beam<br>Width<br>(°) | Nyquist<br>Velocity<br>( <i>m</i> s <sup>-1</sup> ) | Range<br>Resolution<br>(m) | Max<br>Range<br>(km) | Volume<br>scan<br>Interval<br>(min) | Elevations (°)                                       |
| SGDK | 127.43         | 38.11         | 1066                   | 10               | 0.89                 | 64.3                                                | 250                        | 250                  | 10                                  | -0.4 0.0 0.3 0.8 1.4 2.5<br>4.2 7.1 15               |
| SKWK | 126.96         | 37.44         | 615                    | 10               | 0.93                 | 68.3                                                | 250                        | 250                  | 10                                  | -0.2 0.0 0.3 0.8 1.5 2.6<br>4.4 7.3 15               |
| SBRI | 124.62         | 37.96         | 170                    | 10               | 0.96                 | 64.7                                                | 250                        | 250                  | 10                                  | 0.1 0.4 0.8 1.4 2.2 3.4 5.1<br>7.6 15                |
| SKSN | 126.78         | 36.01         | 212                    | 10               | 0.90                 | 67.9                                                | 250                        | 250                  | 10                                  | 0.0 0.3 0.7 1.3 2.1 3.2 5.0<br>7.6 15                |
| CIIA | 126.36         | 37.46         | 142                    | 5                | 0.53                 | 29.7                                                | 250                        | 130                  | ~6                                  | 0.7 1.0 1.5 2.1 2.9 4.0 6.0<br>8.0 11 15 21 28 36 45 |
| CSAN | 126.49         | 36.70         | 45                     | 5                | 0.95                 | 47.9                                                | 250                        | 130                  | 10                                  | 0.5 0.9 1.4 2.0 2.6 3.4 4.5<br>5.9 7.6 10 13 20      |
| XKOU | 127.02         | 37.58         | 136                    | 3                | 0.53                 | 18.0                                                | 60                         | 40                   | 10                                  | 3.0 3.6 4.3 5.1 6.1 7.2 8.6<br>10.2 12.2 14.4 17 20  |
| XYOU | 126.93         | 37.56         | 79                     | 3                | 0.45                 | 18.0                                                | 60                         | 40                   | 10                                  | 2.5 3.0 3.7 4.4 5.4 6.5 7.8<br>9.4 11.4 13.6 16.4 20 |
| XDJK | 126.09         | 37.25         | 116                    | 3                | 1.26                 | 44.8                                                | 150                        | 75                   | 15                                  | 1.5 2.1 3.0 4.2 5.8 7.9 15                           |
| XMIL | 126.44         | 36.93         | 295                    | 3                | 1.26                 | 44.8                                                | 150                        | 75                   | 15                                  | 0.8 1.4 2.2 3.5 5.2 7.9 15                           |
| XSRI | 126.90         | 37.35         | 435                    | 3                | 1.26                 | 44.8                                                | 150                        | 75                   | 15                                  | 0.8 1.4 2.2 3.4 5.2 7.9 15                           |

The radar data were interpolated to the Cartesian coordinate system for WISSDOM synthesis after undergoing quality control (QC). A fuzzy logic QC algorithm was employed to remove non-meteorological signals while preserving useful data (Cho et al., 2006; Ye et al.,

2015). The lowest available radar data (i.e., useful radar reflectivity and radial velocity above ground) were obtained using a relatively high discrimination threshold. Radar data for each there was a topography blockage of more than 10% were removed to retain only realistic data. The radial velocity was unfolded if the radial velocity was folded (i.e., over the Nyquist velocity for each radar). At this stage, the QC radar data had the same grid size as the original coordinates. The the useful and reliable radar data were then interpolated to mitigate possible errors arising from non-meteorological and useless signals in the WISSDOM retrieval (see Section 2.3).

Figure 2 shows the radar coverage and topographic blockage at constant high levels. The mountains are not sufficiently high in South Korea, thus, there were no significant terrain blockages in the WISSDOM domain (Figs. 2a and 2b). In addition, the S-band radars could not provide sufficient observations at lower levels because they were generally located at higher elevations and far from the WISSDOM domain. Although the C-, and X-band radar observations were also limited at the lowest level, they provided good coverage from 0.5 to 1 km MSL (Figs. 2b and 2c). The area of overlap increased from 2 or 3 radars to 5 or 6 radars in the WISSDOM domain below 1 km MSL (mainly due to the short-wavelength radars), then the overlay area was expanded and occupied most areas with 5–7 radar numbers in WISSDOM domain from 2 km, 5 km up to 10 km MSL (Figs. 2d-2f).

Figure 2. (a) The coverage area and topographic blockage of the radar observations were explored at 0.25 km MSL (Mean Sea Level) height, the color shading indicates the overlay areas counting by the radar numbers. The location of S-, C-, and X-band radars were marked by dark blue, light blue, and green triangles, respectively. The black box is the WISSDOM domain as same as in Fig.1a. (b), (c), (d), (e), and (f) are the same as (a), but for the height at 0.5, 1, 2, 5, and 10 km MSL.

An operational sounding at site 47199 (Fig. 1b) collected data every 6 h (from 00Z) each day, and the raising speed was around 3–5 m s<sup>-1</sup> recorded data every 1 s. The sounding observations needed to be interpolated to a fixed vertical spacing of 0.25 km, and temperature profiler was utilized to determine the freezing level, and the horizontal wind information can be used as the background in WISSDOM. The retrieval of horizontal winds (i.e., U- and V-winds) using WISSDOM was evaluated with horizontal winds recorded by the sounding. The dense AWS network measured the surface winds every 1 min within the synthesis domain. Relatively few AWS sites are present over the ocean, but there is a dense distribution overland, especially in Seoul. The AWS observations were also used as background in WISSDOM synthesis. Three RWPs (RWP1–3) were deployed at northeastern and southwestern areas of the synthesis domain (as Fig. 1b). These RWPs provided wind profiles every 50 m from the surface up to 10 km above mean sea level (MSL) at 10 min intervals. The RWPs observations were used as a reference in evaluation of the 3D winds (including W-winds) of WISSDOM.

# 2.2 Overview of the case study

The advantages of using multiple Doppler radars with different wavelengths in WISSDOM were investigated in a frontal squall line case. A short stationary front extending from Shandong Peninsula to Seoul crossed the Yellow Sea at 00Z on 2 August 2020 (Fig. 3a). A nearly stationary subtropical high-pressure system caused this front to occupy the regions in the southeastern ocean off the Korean Peninsula, and a moving low-pressure system moved easterly from 110°E along ~55°N. A local area with high moisture content associated with the low-pressure system eastward also approached Seoul at 12Z on 2 August 2020 (Fig. 3b). Tropical storm Hagupit was also developing in the Pacific Ocean off the eastern coast of Taiwan, and it may be weakly affected the weather systems near South Korea. During this period, a squall line passed Seoul through the WISSDOM domain, and most radars were in operation at this time. This case was selected as an example of a mesoscale convective system that often develops during the warm season in South

Figure 3. Korea Meteorological Administration surface analysis maps obtained at (a) 00:00 UTC and (b) 12:00 UTC on 2 August 2020. The purple shading indicates areas containing high moisture, while the arrows indicate the possible direction of movement. The red circle marked the locations of the Korean Peninsula and the short front.

The evolution of this squall line can be described using the hybrid surface rainfall (HSR, Kwon et al., 2015; Liu et al., 2015; Kwon, 2016). HSR is based on meteorological radar observations that provide high-quality surface rainfall information for South Korea every 10 min (recently, every 5 min) at the lowest height over terrain. The squall line developed with bow-shaped echoes from 03:30 to 06:30 UTC on 2 August 2020 (Figs. 4a–d, respectively). A sharp precipitation gradient was observed along the leading edge, and stratiform precipitation was located behind the convective area. These precipitation structures were typical of a squall line (Houze, 1977; Houze et al., 1989), and broad stratiform areas were present behind a prominent segment of the line as a bow (Fig. 4a). The squall line moved toward Seoul and there were no clear bow-shaped features along the leading edge at 04:30 UTC (i.e., Fig. 4b). Stratiform precipitation developed in the southern segment of the squall line and the bow-shaped characteristics reappeared, but the locations shifted to the southern segment of the squall line,

accompanied by obvious stratiform precipitation areas behind it (Fig. 4c). Compared to the northern segment of the squall line, significant precipitation was observed in its southern segment, and the typical structural characteristics of a squall line were also present. Less organized convection was present in the northern segment of the squall line at 06:30 UTC (Fig. 4d). However, clear bow-shaped structures were recorded in the southern segment when the squall line made landfall. This squall line moved easterly without significant southern or northern movement, with an average moving speed for the leading edge of ~14 m s<sup>-1</sup> from 04:30 to 06:30 UTC.

The performance of WISSDOM wind retrieval was analyzed for this case study at 04:30, 05:30, and 06:30 UTC as the squall line moved from the ocean to the coast and then to the land, respectively. It is also because both clear bow-shaped echoes along the southern segment and

and 06:30 UTC as the squall line moved from the ocean to the coast and then to the land, respectively. It is also because both clear bow-shaped echoes along the southern segment and dissipated bow echoes along the northern segment of the squall line were observed. The characteristics of precipitation and wind patterns (i.e., return flow, etc.) were mainly checked qualitatively before—the accuracy of the retrieved winds was quantified. This step can initially confirm the reliability of retrievals in WISSDOM. Therefore, WISSDOM retrieval could be compared to the typical characteristics of a squall line structure based on Houze et al. (1989). In addition, the squall line was lying over the densest radar network in South Korea at this time, thus observing winds data from a large selection of radars.

Figure 4. Horizontal distribution of the hybrid surface rainfall (HSR) (colored shading, unit: mm  $h^{-1}$ ) at (a) 03:30, (b) 04:30, (c) 05:30, and (d) 06:30 UTC on 2 Aug. 2020.

# 2.3 WISSDOM (WInd Synthesis System using DOppler Measurements)

The first version of WISSDOM was proposed by Liou and Chang (2009) as a mathematical variational-based algorithm used to derive 3D winds using radars and other observations. The basic structure of WISSDOM minimizes the cost function using five constraints (Liou et al., 2012; Tsai et al., 2018, 2022). The cost function can be expressed as eq. (1):

$$J = \sum_{M=1}^{5} J_M, \tag{1}$$

where *M* represents the five constraints. The first constraint is the geometric relations between radar radial velocity and each grid point in WISSDOM using Cartesian coordinates, expressed as follows:

285 
$$J_1 = \sum_{t=1}^{2} \sum_{x,y,z} \sum_{i=1}^{N} \alpha_{1,i} (T_{1,i,t})^2, \qquad (2.1)$$

where t is the time step in Eq. (2.1). WISSDOM uses two time steps. x, y, z indicate the location of the grid points in the synthesis domain, and i is the number (N) of radars.  $\alpha_1$  is the weighting coefficient of  $J_1$ .  $T_{1,i,t}$  is defined using Eq. (2.2):

289 
$$T_{1,i,t} = (V_r)_{i,t} - \frac{\left(x - P_x^i\right)}{r_i} u_t - \frac{\left(y - P_y^i\right)}{r_i} v_t - \frac{\left(z - P_z^i\right)}{r_i} \left(w_t - W_{T,t}\right), \tag{2.2}$$

- where  $(V_r)_{i,t}$  is the radial velocity observed by radar i at time step t,  $P_x^i$ ,  $P_y^i$  and  $P_z^i$  denote
- the coordinate of radar i,  $u_t$ ,  $v_t$  and  $w_t$  ( $W_{T,t}$ ) are the 3D winds (terminal velocity) at a given
- grid point at the time step t.  $r_i$  is defined using eq. (2.3).

$$r_i = \sqrt{(x - P_x^i)^2 + (y - P_y^i)^2 + (z - P_z^i)^2}.$$
 (2.3)

- The second constraint is the difference between the background ( $V_{B,t}$ ) and true wind field
- ( $V_t$ ), which is defined as

$$J_2 = \sum_{t=1}^{2} \sum_{x,y,z} \alpha_2 (\mathbf{V}_t - \mathbf{V}_{B,t})^2,$$
 (3.1)

where  $\alpha_2$  is the weighting coefficient of  $J_2$ , and  $V_t$  is defined as in eq. (3.2):

$$\mathbf{V}_t = u_t \mathbf{i} + v_t \mathbf{j} + w_t \mathbf{k} \,. \tag{3.2}$$

- An anelastic continuity equation, vertical vorticity equation and Laplacian smoothing filter are
- the third, fourth and fifth constraints in eq. (1). They are determined using Eqs. (4), (5), and (6),
- respectively:

$$J_3 = \sum_{t=1}^{2} \sum_{x,y,z} \alpha_3 \left[ \frac{\partial (\rho_0 u_t)}{\partial x} + \frac{\partial (\rho_0 v_t)}{\partial y} + \frac{\partial (\rho_0 w_t)}{\partial z} \right]^2, \tag{4}$$

$$J_4 = \sum_{x,y,z} \alpha_4 \left\{ \frac{\partial \zeta}{\partial t} + \left[ u \frac{\partial \zeta}{\partial x} + v \frac{\partial \zeta}{\partial y} + w \frac{\partial \zeta}{\partial z} + (\zeta + f) \left( \frac{\partial u}{\partial x} + \frac{\partial v}{\partial y} \right) + \left( \frac{\partial w}{\partial x} \frac{\partial v}{\partial y} - \frac{\partial w}{\partial y} \frac{\partial u}{\partial z} \right) \right] \right\}^2, \quad (5)$$

$$J_5 = \sum_{t=1}^{2} \sum_{x,y,z} \alpha_5 [\nabla^2 (u_t + v_t + w_t)]^2, \tag{6}$$

where  $\rho_0$  is the air density, and  $\zeta = \partial v/\partial x - \partial u/\partial y$ .

The WISSDOM domain is presented as the black box in Fig. 1a and in full in Fig. 1b. The domain sizes are 200 × 200 km (10 km) with a spatial resolution of 1 km (0.25 km) in horizontal (vertical). The sounding and AWS observations were adopted as the background constraint for Eq. (3.1). The AWS observations were bilinearly interpolated to the lowest grid point above the ground, and the horizontal distance weighted using a Gaussian distribution between the AWSs and each grid point. Above the surface level, the sounding data provided uniform horizontal winds for each level. The sounding site (#47199) was located at the center of the domain (Fig. 1b) to represent the background of this area. The discrepancies in the retrieved winds were minor when the reanalysis datasets were applied to WISSDOM (not shown), and the results revealed that they were compatible when in-situ storm-scale observations were lacking. Note that the temporal resolution for WISSDOM retrieval was set to every 30 min to synchronize with the radar observations. The basic settings for WISSDOM employed in the present study are summarized in Table 2.

# Table 2. Basic setting for WISSDOM.

| Domain range        | Latitude: 36.545°N-38.344°N<br>Longitude: 125.339°E-27.604°E                            |  |  |  |  |
|---------------------|-----------------------------------------------------------------------------------------|--|--|--|--|
| Domain size         | $200 \times 200 \times 10 \text{ km (length} \times \text{width} \times \text{height)}$ |  |  |  |  |
| Temporal resolution | 30 min                                                                                  |  |  |  |  |
| Spatial resolution  | $1 \times 1 \times 0.25$ km (length × width × height)                                   |  |  |  |  |
| Terrain resolution  | 0.09 km                                                                                 |  |  |  |  |
| Coordinate system   | Cartesian coordinate system                                                             |  |  |  |  |
| Background          | Sounding (#47199) and AWS                                                               |  |  |  |  |

Doppler radars: bilinear interpolation

Background: linear interpolation Data implementation

AWS: bilinear interpolation with Gaussian weighting

Weighting coefficient

Doppler radars :  $\alpha_1 = 10^2$ 

(input datasets) Background  $\alpha_2 = 10^{-1}$ 

One advantage of WISSDOM is that the 3D winds along the radar baseline can be recovered well using a variational-based algorithm. Thus, the quality of the retrieved winds along the radar baseline would not be a significant issue to the radars' relative location (or distance) in WISSDOM, especially when using multiple radars. The other advantage of WISSDOM is that it applies IBM for computing the winds over complex terrain (Liou et al., 2012). The IBM can simulate fluid patterns over a complex geometry in Cartesian coordinates (Peskin, 1972), thus allowing for the extraction of information closer to the surface for each grid in WISSDOM. As it is known that observations are often lacking near the surface, it may be limited to computing and simulating atmospheric variables at the lower boundary, especially over terrain. Therefore, WISSDOM retained and computed the winds from the lowest grid by adopting the IBM, with the retrieved winds better reflecting the real situation at the lower boundary over complex terrain up to higher levels. Those advantages are the reason why SAMURAI was also upgraded by applying the IBM (Bell et al., 2012; Cha et al., 2023), while MUSCAT (Chong and Bousquet, 2001) also uses the IBM, even for on the analysis of tropical cyclones (Cheng et al., 2025).

# 2.4 Scenarios for the use of the radars and corresponding evaluations

Several scenarios were employed in the present study to isolate the contributions of different wavelengths in the radar observations (Table 3). The first three scenarios use only one type of radar in order to determine the impact of different wavelengths individually. The first scenario (scenario S) includes only four S-band radars, and the second and third scenarios employed two C-band and five X-band radars, respectively, and these scenarios (referred to as scenarios C and X, respectively) have not been used in previous WISSDOM analyses. The remaining scenarios

were combinations of radars with different wavelengths. According to previous studies (Liou and Chang, 2009; Liou et al., 2012, 2014, 2016, 2019, 2024; Tsai et al., 2018), S-band radar is necessary in WISSDOM; therefore, the fourth and fifth scenarios combine S-band radars with C-band, and X-band radars, respectively (scenarios SC and SX). Finally, the sixth scenario puts all three radar types together (scenario SCX).

Table 3. List of radars synthesized for each scenario

| Scenarios  | Synthesized Radars                                                                              | Abbreviations |
|------------|-------------------------------------------------------------------------------------------------|---------------|
| Scenario 1 | SKWK, SGDK, SBRI, SKSN (S-band)                                                                 | S             |
| Scenario 2 | CIIA, CSAN (C-band)                                                                             | C             |
| Scenario 3 | XDJK, XMIL, XSRI, XKOU, XYOU (X-band)                                                           | X             |
| Scenario 4 | SKWK, SGDK, SBRI, SKSN (S-band)<br>CIIA, CSAN (C-band)                                          | SC            |
| Scenario 5 | SKWK, SGDK, SBRI, SKSN (S-band)<br>XDJK, XMIL, XSRI, XKOU, XYOU (X-band)                        | SX            |
| Scenario 6 | SKWK, SGDK, SBRI, SKSN (S-band)<br>CIIA, CSAN (C-band)<br>XDJK, XMIL, XSRI, XKOU, XYOU (X-band) | SCX           |

Because the sounding site (#47199) and three radar wind profilers (RWP-1-3) were collocated in the WISSDOM domain, the mean bias (MB) and root mean square deviation (RMSD) between the retrieved WISSDOM winds, soundings, and RWP-1-3 observations were selected as the evaluation metrics in the present study, following the same approach used by Tsai et al. (2023) in evaluating wind retrieval in WISSDOM. Since the vertical spatial resolution of the sounding observations was around 3-5 m, which was associated with the rate of rise of the sensors (3~4 m s<sup>-1</sup>), the data had to be interpolated to 250 m to fit the vertical grid spacing of WISSDOM. The MB and RMSD were estimated by tracking the exact rising path of the sounding sensor because the sounding tracks are not usually right on the grid point of WISSDOM.

Therefore, the sounding observations near the closest grid point in WISSDOM and their retrieved winds were selected to estimate the MB and RMSD. The sounding launching time of 06:00 UTC on 2 August 2020, which was the closest time to the WISSDOM analysis period (05:30 UTC), was selected for this evaluation.

The RWPs were fixed stations that provide vertical 3D wind information from the surface. The RWP observations were interpolated to 250 m to allow for a comparison with the WISSDOM derived winds during the same time steps at 04:30, 05:30, and 06:30 UTC. Similar to the comparison between the sounding observations and the WISSDOM winds, the MB and RMSD were estimated for the RWPs at each site. The MB and RMSD were calculated using Eqs. (7) and (8), respectively:

$$MB = \frac{1}{n} \sum_{i=1}^{n} |(X_i - Y_i)|, \qquad (7)$$

$$RMSD = \sqrt{\frac{\sum_{i=1}^{n} (X_i - Y_i)^2}{n}},$$
 (8)

where n is the number of datapoints, and X and Y represent the observations and synthesized winds, respectively. The vertical profiles for the U- and V-winds from the sounding observations and vertical profiles for the U-, V-, and W-winds from the RWP observations are both compared with the WISSDOM winds for each scenario in Section 3.3, while the MB and RMSD are presented in Section 3.4.

## 3. Results and discussion

# 3.1 Comparison of horizontal wind structure

The precipitation structures and storm-relative flow (considering the movement speed of the squall line at the analysis time) obtained from WISSDOM at 2 km MSL are presented for

scenarios S, C, X, SC, SX, and SCX in Figs. 5a–f, respectively. S-band radars were able to depict clear bow-shaped echoes along the leading edge of the squall line. There were southeasterly and southwesterly winds in advance of and behind the convection region in the southern segment of the line (Fig. 5a). Airflow convergence coincided with this strong convection region. Rear-to-front flow was identified behind the convection region, and the gust front reached  $\sim$ 50 km away from the leading edge of the main squall line, at  $X = \sim 125$  km, as inferred from the weak radar reflectivity areas. The precipitation and airflow structures were similar to typical bow echoes in squall line systems. Along the northern segment of the squall line (i.e., from  $Y = \sim 125$  to Y = 200 km), the convection was relatively weak and less organized. The characteristics of the flow convergence and rear-to-front flow were not clearly detected. Orographic precipitation may have been produced when the winds impinged the mountains near the northeastern area of the synthesis domain.

Compared to scenario S, significant attenuation of radar reflectivity was observed in scenario C (Fig. 5b), particularly in areas where the radar reflectivity was strong. The radar reflectivity was also missing along several azimuths in the northeastern and western sectors relative to the CIIA ( $X = \sim 75 \text{ km}$ ,  $Y = \sim 125 \text{ km}$ ) and CSAN ( $X = \sim 50 \text{ km}$ ,  $Y = \sim 15 \text{ km}$ ) radar sites, due to significant attenuation. Significant flow convergence was also observed coincident with the convection areas along the southern segment of the squall line. Except for the missing reflectivity areas, the airflow structures had characteristics similar to those in scenario S (i.e., rear-to-front flow and flow convergence).

Fig. 5c presents the results from WISSDOM for scenario X. The short detection range of the X-band radars may have reduced the radar reflectivity coverage. The X-band radar reflectivity exhibited greater attenuation compared to scenario S. Furthermore, the X-band radars were concentrated in Seoul (X and Y =  $\sim$ 125 km), so there were no available observations over the ocean near the northwestern corner and the northeastern corner of the synthesis domain. Uniform airflow was observed over regions lacking radar echoes, as the wind information in these areas

was mainly derived from background winds. Although weaker convergence also exists along the convection in the southern segment of the squall line, the rear-to-front flow was unclear. The results indicate high variance in the strength of the radar reflectivity between the long-wavelength (S-band) and short-wavelength radars (C- and X-bands), but the flow structures were similar except for the echo-free areas in scenario X.

Figure 5. Retrieved radar reflectivity (color shading, dBZ), and storm-relative flow (vectors) at 2 km mean sea level (MSL) obtained from WISSDOM for scenarios (a) S, (b) C, (c) X, (d) SC, (e) SX, and (f) SCX. The two black lines indicate the box area corresponding to the mean vertical cross-section A-A' in Fig. 8.

Scenario SC (Fig. 5d) produced almost the same precipitation and storm-relative flow as scenario S (Fig. 5a). Although there were echo-free areas in scenario C (Fig. 5b), the stormrelative flow retained a reasonable structure in scenario SC, especially at the southern end of the squall line (X =  $\sim$ 25–50 km, Y =  $\sim$ 0–25 km). Another flow convergence area coincided with a stronger reflectivity area behind the main convection area near  $X = \sim 0-50$  km,  $Y = \sim 130$  km. Although these signatures were not observed in scenarios S and C, the convergence area was reproduced due to the wider coverage of the C-band radar when combined with some of the S-band radar observations at lower levels (not shown). Scenario SX (Fig. 5e) had minor differences from scenario S, though the results included the observations from the X-band radars. In scenario SCX (Fig. 5f), two distinct flow convergence regions were observed: one along the leading edge of convection in the southern segment of the squall line, and another located behind the convection area, oriented perpendicular to the squall line. The rear-to-front flow exhibited the most prominent bow shape along the squall line. These horizontal airflow and precipitation structures closely matched the typical characteristics of the squall line in mesoscale convective systems (Swastiko et al., 2024) and squall line-like bow echoes in tropical cyclone rainbands (Yu and Tsai, 2013; Yu et al., 2020), meaning that scenario SCX may have produced the most reasonable and representative wind field synthesis.

The W-winds at 2 km MSL for each scenario are presented in Fig. 6. A very clear updraft was found along leading edge and flow convergence areas of the squall line in scenario S (Fig. 6a). A relatively weak updraft was also found in the areas without flow convergence near the areas where the airflow penetrated the leading edge in the northern segment of the squall line (X = ~90 km, Y = ~130 km). W-wind structures are typical of squall lines with downdraft behind and a weak updraft in advance of the convection area. A less clear updraft was captured along the squall line in scenario C (Fig. 6b). However, a stronger updraft core was present in the areas near the center of the synthesis domain. Unclear contrasts between downdrafts and updrafts were present behind and in advance of the convection areas in this scenario. The W-winds in scenario X (Fig. 6c) had no clear relationship with the squall line, with both the updrafts and downdrafts generally weak.

Figure 6. Retrieved vertical velocity (i.e., W-winds, color shading, m s<sup>-1</sup>), and storm-relative flow (vectors) at 2 km MSL obtained from WISSDOM for scenarios (a) S, (b) C, (c) X, (d) SC, (e) SX, and (f) SCX. The two black lines indicate the box area corresponding to the mean vertical cross-section A-A' in Fig. 9.

However, a prominent updraft was produced along the squall line in scenarios SC and SX (Figs. 6d and 6e). In addition, the updraft areas were expanded in advance of the leading edge and behind the gust front in the southern segment of the squall line. These expanded updraft areas became clearer in scenario SCX (Fig. 6f), revealing a stronger updraft in these areas. A clear updraft was present along the convection of the squall line, and a stronger downdraft was also seen behind the convection areas coincident with the rear-to-front flow.

# 3.2 Comparison of vertical wind structure

Because the precipitation and storm-relative flow in the southern segment of the study squall line were very similar to the typical structure of a squall line (Fig. 7; Houze et al., 1989), the present study analyzed the average precipitation and flow structure in the southern segment of

the squall line. The averaged cross-section is indicated by A-A' in Fig. 5a. The retrieval results could then be compared to the reference for a typical squall line.

Figure 7. Conceptual model of a cross-section perpendicular to the orientation of the squall line, The thick solid line and grey-shaded areas indicate the precipitation echoes observed from the radar (adopted from Figure 1 in Houze et al., 1989).

Precipitation and flow structures from scenario S (Fig. 8a) closely resembled those of a typical squall line (Fig. 7), using a radar echo threshold of 25 dBZ, because the intense precipitation and significant flow structures could be successfully identified in this case. The strongest updraft was associated with heavy precipitation areas and descending rear-to-front inflow behind the convection with the stronger radar reflectivity. The descending rear-to-front inflow appeared to be a return flow that descended to near the surface; however, the return flow could not be clearly seen, which may have been caused by the lack of data at lower levels. The gust front was also detected in scenario S, with a weak updraft just above it. Although the attenuation produced weaker radar reflectivity in the convection areas in scenario C, storm-relative flow was observed (i.e., the environmental wind subtracted from the moving speed of the precipitation systems, Fig. 8b). Unlike scenario S, return flow could not be produced in scenario C. However, the C-band radars produced more radar observations near the surface (cf. Fig. 2). A weak updraft and lack of descending rear-to-front inflow were the main characteristics of scenario X (Fig. 8c). Nevertheless, the X-band radars were the same as C-band radars in that they provided more radar observations at lower levels. Note that the front-to-rear flow could only be retrieved

near the surface ( $\sim$ 0.5 km MSL) in scenario X, and this characteristic was similar to a typical squall line (Fig. 7).

Figure 8. Mean cross-section of the retrieved radar reflectivity (color shading, dBZ), and storm-relative flow (vectors) obtained from WISSDOM for scenarios (a) S, (b) C, (c) X, (d) SC, (e) SX, and (f) SCX corresponding to the A-A' box in Fig. 5a.

The precipitation and flow structures were similar between scenarios S, SC, and SX (Figs. 8a, 8d, and 8e). However, the C- and X-band radars provided sufficient radar observations near the surface, thus the descending rear-to-front inflow appeared to return at very low levels near the surface. In scenario SCX (Fig. 8f), a strong updraft was associated with strong radar reflectivity in the convection areas of the squall line. In addition, another updraft was observed coincident with the gust front and above it (i.e., the position of the new cell indicated in Fig. 7). Furthermore, descending rear-to-front inflow occurred behind the convection area, and this inflow changed to be the return flow near the surface. Although the C- and X-band radars experienced significant attenuations, adding S-band radar observations can compensated for this. Similarly, although S-band radars lack of observations at lower levels, this weakness was minimized by adding C- and X-band radar observations in scenario SCX. Overall, the results

# derived from WISSDOM synthesis were comparable to the characteristics of a typical squall line.

Figure 9. The same as Fig. 8, but for a mean cross-section of the vertical velocity (i.e., W-winds, color shading, m  $s^{-1}$ ) and storm-relative flow (vectors) obtained from WISSDOM for scenarios (a) S, (b) C, (c) X, (d) SC, (e) SX, and (f) SCX corresponding to the A-A' box in Fig. 6.

The variance in the intensity of the W-component is presented for each scenario in Fig. 9. Only one updraft core (defined as a vertical velocity over 1.5 m s<sup>-1</sup> with upward extension at least 5 km tall, marked in dark orange color) was presented in scenario S (Fig. 9a), while there were two updraft cores in scenario C (Fig. 9b). The second updraft core was just located above the areas from the leading edge of the squall line to the gust front. This updraft plays a role in generating new cells in the squall line, and this updraft can also be found in a typical squall line (cf. Fig. 7). There was no clear updraft in scenario X (Fig. 9c), but positive values for the W-component were retrieved in the convection of areas of the squall line. The intensity of the updraft cores was stronger in scenario SC (Fig. 9d), while only one updraft core was present in scenario SX (Fig. 9e). Figure 9f shows that two updraft cores were observed in scenario SCX, and an intense downdraft was presented in behind one of the updraft cores in the convection areas. These results had characteristics similar to a typical squall line in this case, thus

highlighting the positive impact of adding C- and X-band radar observations to S-band radars as they can provide sufficient data at lower levels.

#### 3.3 Quantitative evaluation of retrieved winds

The results from WISSDOM were able to qualitatively describe the precipitation and flow structures, but the quantitative accuracy of the retrieval winds required further verification. The optimal scenario for WISSDOM also needed to be identified by running a series of evaluations. In the present study, the performance of WISSDOM was evaluated against the sounding and RWP data. Since the sounding continuously ascended, the WISSDOM winds were extracted by following the trajectories of the soundings. 10a presents the U-winds profiles from both the sounding observations and the various WISSDOM scenarios. Below 4 km MSL, the differences between the sounding observations and the WISSDOM-retrieved winds were minimal. However, above 4 km MSL, the WISSDOM winds deviated from the sounding observations, as wind speeds dropped significantly near 5 km MSL. Above ~6 km MSL, the sounding observations and WISSDOM winds once again showed good agreement. The WISSDOM winds were consistent for each scenario except scenarios C and scenario X, coinciding with the changes in the sounding winds at  $\sim$ 5 km MSL. The differences in the V-winds between the sounding observations and WISSDOM synthesis winds are presented in Fig. 10b. Overall, the results indicate minor differences, except that scenario X produced higher V-wind speeds than the sounding observations below ~5 km MSL. The overall performance of WISSDOM in retrieving the winds was good despite the abrupt changes in the sounding wind speeds at certain levels in this case. Note that scenario SCX had relatively smooth trends, without significant fluctuations to changes in the sounding observations. The more consistent results obtained from the different scenarios in WISSDOM synthesis may

be related to the sufficient coverage of the radar observations because the sounding was launched

near the center of the synthesis domain.

Figure 10. Vertical profiles of (a) the U-winds and (b) V-winds observed at sounding site #47199 (thick black line) at 06:00 UTC on 2 August 2020. Thin lines with numbers and colors indicate different scenarios. Number 1 colored black indicates scenario S (see Table 3). Numbers 2, 3, 4, 5, and 6 colored red, blue, green, pink, and orange indicate the scenarios C, X, SC, SX, and SCX, respectively.

The RWPs provided the average vertical profiles of U-winds, V-winds, and W-winds, allowing the WISSDOM winds to be compared above these three RWPs during the three stages from 04:30 to 06:30 UTC on 2 August 2020. Figure 11 describes the differences between the WISSDOM winds and three RWPs. The U-winds in scenario SCX exhibited the smallest differences compared to RWP1 (Fig. 11a) except for heights below ~1.5 km MSL. The U-winds in scenario X more closely resembled RWP1 at lower levels, but there were more significant differences between ~1.5 and 8 km MSL. The V-winds in scenario SCX also had the smallest differences from RWP1 (Fig. 11b) but only below ~6 km MSL. In contrast, the results were the opposite for scenarios SCX and X, with the V-winds in scenario X exhibiting the least significant difference compared to RWP1 above ~6 km MSL but a more significant difference below ~6 km MSL. A relatively more significant updraft was detected by RWP1 below ~5 km MSL (Fig. 11c), and all scenarios produced significant differences from the W-winds of RWP1 at these levels.

Figure 11. (a) Average vertical profiles of the U-wind speed (thick black line) observed at RWP1 at 04:30, 05:30, and 06:30 UTC on 2 August 2020. The thin lines with numbers and colors indicate different scenarios. Number 1 colored black indicates scenario S (see Table 3). Numbers 2, 3, 4, 5, and 6 colored red, blue, green, pink, and orange indicate the scenarios C, X, SC, SX, and SCX, respectively. (b), (c) The same as (a) but for V-winds and W-winds. (d), (e) and (f) are the same as (a), (b), and (c) but for RWP2. Note that only two time steps (04:30 and 05:30 UTC) were included in (f). (g), (h) and (i) are the same as (a), (b), and (c) but for RWP3.

Although observations from RWP2 were missing and smaller in the mid-levels, the U-, V-, and W-winds could still be compared with WISSDOM winds (Figs. 11d–f). There were similar trends and smaller differences between RWP2 and each scenario, with the most obvious differences occurring near the mid-levels, though they were  $\sim$ 5 m s<sup>-1</sup>. In particular, the V-winds

observed by RWP2 exhibited minor differences from every WISSDOM scenario. RWP2 observed a relatively weak downdraft, while the W-winds from WISSDOM were weak below ~4 km MSL. Smaller differences were found above 6 km MSL of only ~0.5 m s<sup>-1</sup>, though RWP-2 W-winds were not included at 06:30 UTC due to missing data. The U-winds produced in scenario X had obvious differences from the other scenarios and the RWP3 observations (Fig. 11g). Although RWP3 lacked data above 6 km MSL, it exhibited similar trends and values for U-winds in comparison to the WISSDOM winds. There were differences in V-winds at around 10 m s<sup>-1</sup> between the RWP3 observations and the WISSDOM winds (Fig. 11h) except for scenario X (~20 m s<sup>-1</sup>). It is important to note that the quality of the W-winds observed by RWP3 was not completely reasonable because an updraft with values exceeding 6 m s<sup>-1</sup> was observed only at ~4 km MSL. Therefore, the W-wind observations from RWP3 were not used to evaluate the WISSDOM winds in the present study. Nevertheless, the WISSDOM winds produced more reasonable results, with the downdraft observed behind the squall line near the RWP3 site (Figs. 1b and 5). The MB and RMSD for the comparison between the sounding and RWP observations and the WISSDOM winds for each scenario are presented in Fig. 12. The MB for the horizontal winds is displayed in Fig. 12a. The MB for the U-winds and V-winds was 1 m s<sup>-1</sup> between the sounding observations and every WISSDOM scenario (thin black lines). A larger MB was produced at RWP1 for the U- and V-winds of around 1 m s<sup>-1</sup> and 3.5 m s<sup>-1</sup>, respectively, between each scenario (red lines). The MB for the horizontal wind speeds was  $\sim 3.5 \text{ m s}^{-1}$  between the RWP2 observations and every WISSDOM scenario (green lines). The MB values were observed for RWP3 (less than 2 m s<sup>-1</sup>) for each scenario, with a maximum MB for the U-winds of 1.6 m s<sup>-1</sup> in scenario S and for the V-winds of more than 3 m s<sup>-1</sup> for scenario X (blue lines). Although the lowest mean MB for the horizontal winds (i.e., counting U-winds and V-winds) was 0.93 m s<sup>-1</sup> for scenario C (the thick black line in Fig. 12a), a slightly higher of mean MB (1.01 m s<sup>-1</sup>) was observed between the observations and scenario SCX. The MB for W-winds was also low at around  $-0.5 \text{ m s}^{-1}$  between

RWP2 and every WISSDOM scenario (the green line in Fig. 12b). However, the MB for the W-winds ranged between  $\sim 2.5$  m s<sup>-1</sup> in the comparison between RWP1 and the WISSDOM scenarios (the red line in Fig. 12b), and the lowest mean MB for the W-winds was 1.1 m s<sup>-1</sup> for scenario SCX (the thick black line in Fig. 12b).

Figure 12. (a) Mean bias (MB) of the U-wind speed (solid lines marked with U) and V-wind speed (dashed lines marked with V) for every scenario in WISSDOM and for the sounding (black lines marked with S), RWP1 (red lines marked with 1), RWP2 (green lines marked with 2), and RWP3 (blue lines marked with 3) data. The thick black line indicates the mean MB of U-winds and V-winds. (b) The same as (a) but for W-wind speed (solid lines marked with W) and mean MB of W-winds. (c) The same as (a) but for the root mean square difference (RMSD), but The thick black line indicates the mean RMSD of U-winds and V-winds. (d) The same as (c) but for the W-wind speed (solid lines marked with W).

The RMSD for the horizontal winds is presented in Fig. 12c. The RMSD for the U- and Vwinds was around 1.7 m s<sup>-1</sup> when comparing the sounding observations with each WISSDOM scenario (thin black lines), whereas an RMSD for the horizontal wind speed was ~2–4 m s<sup>-1</sup> based on the RWP2 observations (green lines). However, the RMSD for the horizontal winds at RWP1 (red lines) and RWP3 (blue lines) varied widely across the WISSDOM scenarios, ranging from  $\sim$ 2 m s<sup>-1</sup> to 9 m s<sup>-1</sup>. The overall RMSD for the horizontal winds was suitably low in scenario SCX, even at RWP1 (less than ~4 m s<sup>-1</sup>) and RWP3 (~5 m s<sup>-1</sup>). The lowest mean MB for the horizontal winds was 1.57 m s<sup>-1</sup> for scenario SCX (the thick black line in Fig. 12c). Fig. 12d presents the RMSD for the W-winds between RWP1 and RWP2. The RMSD was ~0.7 m s<sup>-1</sup> and ~2.5–3.0 m s<sup>-1</sup> at RWP2 and RWP1, respectively, in comparison with the WISSDOM scenarios. The lowest mean MB for the W-winds was 1.5 m s<sup>-1</sup> for scenario SCX (the thick black line in Fig. 12d). The mean MB and RMSD values in the comparison between the sounding observations and average statistic values of three RWPs (if any) and WISSDOM scenarios are summarized in Table 4. Overall, scenario SCX produced lower MB and RMSD values than the other scenarios, indicating that the performance of WISSDOM can be improved by adding C- and X-band radar observations. Note that because the verification observations are being used in the WISSDOM synthesis, the results of the sounding observations are not verified independently (Tsai et al., 2023); nevertheless, this present study mainly documented the variances of each scenario and potential errors of retrieval winds from the WISSDOM.

Table 4. Comparisons between the sounding and RWPs for each scenario during 04:30 and 621 06:30 UTC on 2 August 2020.

|   | Mea        | nn Bias (MB, m s | <sup>-1</sup> ) | Root Mean Square Difference (RMSD, m s <sup>-1</sup> ) |           |         |  |
|---|------------|------------------|-----------------|--------------------------------------------------------|-----------|---------|--|
|   | U-winds    | V-winds          | W-winds         | U-winds                                                | V-winds   | W-winds |  |
| S | 0.1 / 1.6* | 0.2 / 2.6        | —/ 1.3          | 1.6 / 3.5                                              | 1.6 / 4.1 | — / 1.7 |  |
| C | 1.2 / 1.4  | 1.1 / 1.6        | —/ 1.3          | 2.5 / 3.4                                              | 1.6 / 3.6 | — / 1.6 |  |
| X | 0.8 / 0.9  | 0.8 / 2.6        | — / 1.5         | 1.5 / 4.5                                              | 2.1 / 4.5 | — / 1.6 |  |

| SC  | 0.6 / 1.2 | 0.7 / 2.1 | — / 1.2         | 1.7 / 3.2 | 1.7 / 4.0 | <u> </u> |
|-----|-----------|-----------|-----------------|-----------|-----------|----------|
| SX  | 0.2 / 1.5 | 0.2 / 2.6 | —/ 1.3          | 1.5 / 3.6 | 1.6 / 4.2 | <u> </u> |
| SCX | 0.7 / 1.4 | 0.5 / 2.0 | — / 1. <b>0</b> | 1.7 / 3.1 | 1.7 / 3.9 | / 1.5    |

\*Sounding / RWPs

#### 3.4 Discussions

WISSDOM typically employs multiple S-band radar observations, sometimes supplemented with one or two additional short-wavelength C-band or X-band radars. The present study thus aimed to quantify the contributions of S-, C- and X-band radars in WISSDOM in terms of radar reflectivity, U-winds, V-winds, and W-winds. To clarify this, the horizontal and vertical differences between scenario S and scenario SCX are presented in Figs. 13 and 14, respectively. The differences in the radar reflectivity between scenarios S and SCX were relatively minor (±5 dBZ) (Fig. 13a) except for a larger difference (> 15 dBZ) over the mountainous areas (i.e., the northeastern part of the synthesis domain). These characteristics reveals typical squall line as most precipitation areas were located behind the leading edge. It is possible that the S-band radars could not cover lower levels because they are located at high altitudes or that the terrains blocked the C-band and X-band radars due to the lower altitude of the radar sites. Strong positive U-winds  $(\sim 3-9 \text{ m s}^{-1})$  appeared behind the convection areas of the squall line, while negative U-winds (< 6 m s<sup>-1</sup>) were observed in the areas in the southeastern region of the synthesis domain (Fig. 13b). This means that incorporating the short- wavelength radars enhances both rear-to-front and frontto-front flow structures. These results were also consist with typical squall line as stronger rearto-front flow can be found in this case.

A second convergence area was detected in between the northern and southern segments of the squall line, with obviously negative (> 15 m s<sup>-1</sup>) and positive V-winds present in Fig. 13c (X =  $\sim$ 0–75 km, Y =  $\sim$ 100–150 km). Positive V-winds also penetrated the northern segment of the squall line, which could be explained by the less organized precipitation structures in this region.

These results indicate that the short-wavelength radars provided detailed wind information for WISSDOM analysis. Significantly positive W-winds differences (> 3.5 m s<sup>-1</sup>) were present in advance of the squall line extending to the gust front (Fig. 13d). Incorporating short-wavelength radars observations resulted in a noticeable increase in the overall differences in W-winds. The results reasonable reproduced stronger updraft along the leading edge of squall line.

Figure 13. (a) The difference in the radar reflectivity between scenarios SCX and S (S is subtracted from SCX) at 2 km MSL. (b), (c) and (d) are the same as (a), but for U-, V-, and W-winds, respectively.

Differences in the average radar reflectivity along the A-A' cross-section are displayed in Fig. 14a. Most of the positive radar reflectivity differences were present below 1 km MSL behind the convection area of the squall line. The maximum positive radar reflectivity differences were observed at around X = 75 km (> 35 dBZ), coinciding with the strong convection of the squall line. The short-wavelength radars thus provided important observations at lower levels for the WISSDOM analysis. Fig. 14b revealed significant positive U-winds differences ( $\sim 3-15$  m s<sup>-1</sup>) behind the squall line from  $\sim 3$  km MSL down to the ground. The real-to-front flow was intensified by adding the short-wavelength radar observations. Consequently, while the U-wind component exhibited substantial changes, the V-winds differences behind the squall line remained minor (Fig. 14c), suggesting that the short-wavelength radar observations had little impact on the V-wind component in that region. Positive W-winds differences ( $\sim 1-2$  m s<sup>-1</sup>) were found in advance of the squall line up to the boundary of the gust front (Fig. 14d). The short-wavelength radars thus resolved the updraft above the gust front where new cells were generated.

The precipitation and kinematic structures of the scenario SCX were most similar to a typical squall line (cf. Figs. 7, 8f, and 9f). The performance of the scenario SCX was also quantitatively evaluated (cf. Fig. 12), with the results indicating that the optimal scenario used a larger number of radars spanning multiple wavelengths, including the S-, C-, and X-band radars. Although the S-band radar can provide good coverage of radar reflectivity without obvious attenuations, the precipitation and radial velocity information were usually missed at lower layers because of the high altitude of the radar sites. The C- and X-band radars were characterized by significant attenuations but still provided sufficient radial velocity information, especially in the lower layers. In WISSDOM, the availability of additional data improves the accuracy of the retrieval for low-level boundary conditions. Thus, the C- and X-band radars are essential in WISSDOM synthesis for more accurate 3D wind retrieval if they can cover more lower-level areas. Based on

the setup, it was beneficial in this case study, however, the performance of WISSDOM retrieval requires further evaluation using other cases and weather phenomena.

Figure 14. Same as Fig. 13 but for the average cross-section corresponding to the box along A-A' in Fig. 13.

#### 4. Conclusion

This study first employed 11 radars in WISSDOM to retrieve 3D winds from a squall line system that passed Seoul, South Korea, at 05:30 UTC on 2 August 2020. Different scenarios were established (cf. Table 2) to identify the differences between radars with different wavelengths when adopted in WISSDOM. The advantages of combining the four S-band, two C-band, and five X-band radars were documented, and the performance of each scenario was evaluated.

Based on the results of this study, the four S-band radars provided good radar reflectivity and radial velocity with sufficient coverage and without attenuation (cf. Fig. 5a). However, there

were no available observations below ~1 km MSL due to the high altitude of the radar sites (cf. Table 1). Although the two C-band and five X-band radars experienced significant attenuation, they were able to fill the observation gaps for the S-band radars near the surface. The more complete observations allowed for the retrieval of high-quality winds from WISSDOM because their lower boundary conditions could be more accurately described. Scenario SCX produced structures similar to those of a typical squall line. Thus, a more substantial rear-to-front flow and a stronger updraft were found in scenario SCX, highlighting the importance of adding short-wavelength radars to WISSDOM.

The performance of each scenario was quantitatively evaluated using the MB and RMSD between the sounding observations, RWPs, and 3D winds retrieved by WISSDOM. The MB for the U- and V-winds between the sounding observations and scenario SCX were -0.7 and 0.5 m s<sup>-1</sup>, respectively, while the RMSD was 1.7 m s<sup>-1</sup> for both components. Similarly, the average MB was -0.1, 0.2, and 0.6 m s<sup>-1</sup> and the RMSD was 2.3, 3.6, and 1.2 m s<sup>-1</sup> for the U-, V-, and W-winds, respectively, when comparing the WISSDOM retrieval results and the three RWP observations (Table 4). These results indicate that the scenario SCX was the optimal and most stable configuration, though there were differences between the retrieved WISSDOM winds and the RWP observations near the margins of the synthesis domain.

This study suggests that a network of radars operating at multiple wavelengths can be used to derive high-quality 3D winds using WISSDOM for severe weather systems such as squall lines. Although the results are positive in this case study, the configuration of WISSDOM retrievals may vary case by case. This finding is a great step forward but has only been tested in a squall line-type system, geographically positioned so the current network and WISSDOM configuration has a positive result, but that for other cases, that configuration might change. In the future, other weather systems such as typhoons and fronts can be included in the analysis. Furthermore, the effect of combining radars in other wind retrieval algorithms such as SAMURAI and MUSCAT should also be documented, while more 3D wind observations are required to

verify the performance of these algorithms. In addition, the impact of severe weather needs to be clearly understood in order to prevent disasters, for which optimizing the performance of WISSDOM holds great importance.

- Code and data availability. The radar, sounding, radar wind profiler, HSR, WISSDOM and
- AWS dataset is freely available from the KMA website (https://data.kma.go.kr). Please note
- that the official language of this website is Korean, and more information and assistance can be
- found in their interface when proceed with the registration
- (https://data.kma.go.kr/cmmn/selectMemberAgree.do). Figures were made with NCL (NCAR
- Command Language) version 6.2.2 (http://dx.doi.org/10.5065/D6WD3XH5).

724

- Acknowledgments. This study is supported by the National Science and Technology Council of
- Taiwan under Grants NSTC113-2111-M-034-005. This work was funded by the Korea
- Meteorological Administration Research and Development Program "Observing Severe Weather
- in Seoul Metropolitan Area and Developing Its Application Technology for Forecasts" under
- Grant (KMA2018-00125). The authors would like to thank the participants of the field campaign
- "Korea Precipitation Observation Program: international collaborative experiments for
- Mesoscale convective system in Seoul metropolitan area" (KPOP-MS), hosted by the Korea
- Meteorological Administration (KMA). We acknowledge the critical comments from anonymous
- reviewers and editor. Thanks to Ms. Chia-Jing Wu for plotting the figures.

- Author contributions. This work was made possible by contribution from all authors.
- Conceptualization, CLT, GWL; methodology, CLT, YCL, and KK; software, CLT, YCL, and
- KK; validation, KK, YCL, and GWL; formal analysis, CLT, and KK; investigation, CLT, and
- GWL; writing—original draft preparation, CLT; writing—review and editing, GWL, YCL and
- KK; visualization, CLT; supervision, GWL; funding acquisition, CLT and GWL. All authors
- have read and agreed to the published version of the manuscript.

741742

Competing interests. The authors declare that they have no conflict of interest.

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

| 871 | Tseng, Y., and Ferzige     | r, J.: A ghost         | -cell immersed bo    | undary method for                                    | flow in complex            |
|-----|----------------------------|------------------------|----------------------|------------------------------------------------------|----------------------------|
| 872 | geometry.                  | J.                     | Comput.              | Phys.,                                               | <b>192</b> , 593–          |
| 873 | 623, <u>https://do</u>     | i.org/10.1016/j.       | jcp.2003.07.024, 2   | 003.                                                 |                            |
| 874 | Varble, A., Zipser, E. J., | Fridlind, A. M.,       | Zhu, P., Ackerman    | n, A. S., Chaboureau                                 | , JP., Collis, S.,         |
| 875 | Fan, J., Hill, A., a       | and Shipway, B         | .: Evaluation of clo | oud-resolving and lin                                | nited area model           |
| 876 | intercomparison s          | simulations usir       | ng TWP-ICE obser     | rvations. 1: Deep co                                 | nvective updraft           |
| 877 | properties, J. Geo         | phys. Res., <b>119</b> | , 13891–13918, 20    | 14.                                                  |                            |
| 878 | Ye, BY., Lee, G. and Pa    | ark, HM.: Ider         | ntification and remo | oval of non-meteoro                                  | logical echoes in          |
| 879 | dual-polarization          | radar data basec       | d on a fuzzy logic a | lgorithm. Adv. Atmo                                  | s. <i>Sci.</i> , 32, 1217– |
| 880 | 1230, https://doi.o        | rg/10.1007/s003        | 76-015-4092-0.,201   | <u>5</u> .                                           |                            |
| 881 | Yu, C K., Cheng, L         | W., Wu, C C.           | , and Tsai, C L.:    | Outer Tropical Cy                                    | clone Rainbands            |
| 882 | Associated with T          | Typhoon Matmo          | o (2014). Mon. Wes   | a. Rev., <b>148</b> , 2935–2                         | 952                        |
| 883 | https://doi.org/10.        | .1175/MWR-D-           | -20-0054.1, 2020.    |                                                      |                            |
| 884 | Yu, C K., and Tsai, C      | L.: Structural a       | nd surface features  | of arc-shaped radar                                  | echoes along an            |
| 885 | outer tropical cyc         | lone rainband.         | J. Atmos. Sci., 70,  | 56–72, <a href="https://doi.org">https://doi.org</a> | g/10.1175/JAS-D-           |
| 886 | <u>12-090.1</u> ., 2013.   |                        |                      |                                                      |                            |