# Peer review of "(untitled)"

_EGUsphere, 2025_

## Author Comment (AC2)

**Review of "Advantages of using multiple Doppler radars with different wavelengths for three-dimensional wind retrieval." By Tsai et al.**

This paper provides an overview of multi-Doppler analyses of a bow-echo that passed through South Korea on 2 August 2020. This convective system was sampled by a network of 11 radars of varying wavelengths. The authors conduct experiments where variational wind retrievals are made using only specific wavelength radars for given field experiments. While the authors conduct an exercise that would have potentially useful implications for how wind retrievals are calculated, there are numerous problems that prevent me from recommending this paper for publication.

We appreciate that Referee#1 provided helpful and insightful comments, which helped us substantially improve the manuscript. We have carefully checked the comments, and the context of the scanning strategies of the radars; statistical analyses and quantitative evaluations for earlier and late stages of WISSDOM synthesis have been added to this revision. In addition, we have emphasized the benefits of WISSDOM for recovering the winds along the radar baseline. The figures were modified in a new color setting, and the updraft code can be identified easily. A set of responses to the comments is provided below. Specific locations of modified portions (marked as underlines) were also noted as the number of lines in the revised manuscript.

**Fatal flaws:**
For one, the results are not placed in the context of the scanning strategies of the radars. Were the radars on a synchronous scan strategy? In addition, The S-band radars are all placed relatively close together, while the C and X band radars are further out, making a more optimal baseline for multiple Doppler retrievals. Could this also be a factor as well?

1. Thank you for the comments, the authors have provided more information related to the scanning strategies of the radars in the revision. The radars used in this study were operated in temporal resolution by around 6~15 min with the PPI (plan position indicator) elevations from −0.4° and 20° (45° for CIIA). The variances of these scanning strategies were because they were operate by different departments from the governments and universities.  However, these radars can be synchronized in every 30 min time window because they have the similar purposes for detecting severe

weather systems in this region. The details of temporal resolution, PPI elevation angles, and basic setting for each radar were revised in the Table 1. The authors also have emphasized that the synchronized scanning strategies were applied appropriately in WISSDOM for wind retrievals in the revision:

L168-174: "The temporal resolution for each radar volume scan was 10 min except for CIIA (XDJK, XMIL, and XSRI), which was around 6-7 min (15 min), the complete volume scan can be synchronized every 30 min from the selected radars. In a complete volume scan of each radar, the PPI (plan position indicator) elevation angles were between −0.4° and 20° (45° for CIIA), the details of the elevation angles can be found in Table 1. Fundamentally, the radars used in this study are mostly synchronized in similar scanning strategies, even though they were operated from different departments of governments and universities.".

L183: Table 1. Specifications for the radars used in the present study

| | Longitude (°E) | Latitude (°N) | Radar Height (m) | Wave Length (cm) | Beam Width (°) | Nyquist Velocity (m s⁻¹) | Range Resolution (m) | Max Range (km) | Time Interval (min) | Elevations (°) |
|---|---|---|---|---|---|---|---|---|---|---|
| SGDK | 127.43 | 38.11 | 1066 | 10 | 0.89 | 64.3 | 250 | 250 | 10 | −0.4 0.0 0.3 0.8 1.4 2.5 4.2 7.1 15 |
| SKWK | 126.96 | 37.44 | 615 | 10 | 0.93 | 68.3 | 250 | 250 | 10 | −0.2 0.0 0.3 0.8 1.5 2.6 4.4 7.3 15 |
| SBRI | 124.62 | 37.96 | 170 | 10 | 0.96 | 64.7 | 250 | 250 | 10 | 0.1 0.4 0.8 1.4 2.2 3.4 5.1 7.6 15 |
| SKSN | 126.78 | 36.01 | 212 | 10 | 0.90 | 67.9 | 250 | 250 | 10 | 0.0 0.3 0.7 1.3 2.1 3.2 5.0 7.6 15 |
| CIIA | 126.36 | 37.46 | 142 | 5 | 0.53 | 29.7 | 250 | 130 | ~6 | 1.4 1.9 2.5 3.2 4.0 5.0 7.0 10 15 20 26 32 38 45 |
| CSAN | 126.49 | 36.70 | 45 | 5 | 0.95 | 47.9 | 250 | 130 | 10 | 0.5 0.9 1.4 2.0 2.6 3.4 4.5 5.9 7.6 10 13 20 |
| XKOU | 127.02 | 37.58 | 136 | 3 | 0.53 | 18.0 | 60 | 40 | 10 | 3.0 3.6 4.3 5.1 6.1 7.2 8.6 10.2 12.2 14.4 17 20 |
| XYOU | 126.93 | 37.56 | 79 | 3 | 0.45 | 18.0 | 60 | 40 | 10 | 2.5 3.0 3.7 4.4 5.4 6.5 7.8 9.4 11.4 13.6 16.4 20 |
| XDJK | 126.09 | 37.25 | 116 | 3 | 1.26 | 44.8 | 150 | 75 | 15 | 1.5 2.1 3.0 4.2 5.8 7.9 15 |
| XMIL | 126.44 | 36.93 | 295 | 3 | 1.26 | 44.8 | 150 | 75 | 15 | 0.8 1.4 2.2 3.5 5.2 7.9 15 |
| XSRI | 126.90 | 37.35 | 435 | 3 | 1.26 | 44.8 | 150 | 75 | 15 | 0.8 1.4 2.2 3.4 5.2 7.9 15 |

L312-313: "Note that the temporal resolution of WISSDOM retrieval was set to every 30 min to synchronize with radar observations.".

2.  Yes, the S-band radar can detect data far from the radar rather than the C- and X-band radars (cf. Max Range in Table 1). Since WISSDOM is a variational-based approach to derive the 3D winds, the wind fields can be recovered well along the radar baseline [Please see L92: One of the advantages of this approach is that winds can be recovered along the radar baseline, and high-quality winds can also be derived over complex terrain (Liou et al., 2012, 2013, 2014, 2016; Lee et al., 2018)].

    Thus, the quality of retrieved winds along the radar baseline was not affected (or minor) with the location and distance of the radars. Based on these, your concern (the S-band is placed relatively closer, and C- and X-band radars are further out), which are not the key factors affecting the retrieved winds along the radar baseline in WISSDOM. The authors have explained and emphasized these statements in the revision: L316-319: "One advantage of WISSDOM is that the 3D winds along the radar baseline can be recovered well using a variational-based algorithm. Thus, the quality of retrieved winds along the radar baseline would not be a significant issue to the radars' relative location (or distance) in WISSDOM, especially when using multiple radars.".

In the analysis of the updraft cores I found it hard to determine the number of updraft cores simply by eye. Have the authors considered counting these using thresholding techniques (see Varble et al (2014?)). Finally, The MB and RMSD values in the quantitative analysis in Table 4 do not clearly favor the SCX regime, and do not seem to demonstrate any quantitative improvement of using SCX over just S.  Do the authors have statistics for earlier and later stages of this storm, or other cases, that would provide a larger amount of data for analysis?

1.  Thanks for pointing out this problem. The color bar has been adjusted in Figures 6 and 9, in the manuscript (L436 and 489). The number of updraft cores can be easily identified from the figures, and relatively stronger updraft was emphasized by the dark orange and red colors from 0.9 to 3 m s$^{-1}$ . The revised figures are also shown below.

[Figure]

Figure 6. Retrieved vertical velocity (i.e., W-winds, color shading, m s⁻¹), and storm-relative flow (vectors) at 2 km MSL obtained from WISSDOM for scenarios (a) S, (b) C, (c) X, (d) SC, (e) SX, and (f) SCX. The two black lines indicate the box area corresponding to the mean vertical cross-section A-A' in Fig. 9.

[Figure]

Figure 9. The same as Fig. 8, but for a mean cross-section of the vertical velocity (i.e., W-winds, color shading, m s⁻¹) and storm-relative flow (vectors) obtained from WISSDOM for scenarios (a) S, (b) C, (c) X, (d) SC, (e) SX, and (f) SCX corresponding to the A-A' box in Fig. 6.

2. The authors do not consider counting the vertical velocity using thresholding techniques (Collis et al., 2013; Varble et al., 2014) because WISSDOM has considered the terrain features with IBM and adopted the vorticity equation to be one of the constraints, which can supposedly improve the quality of retrieval wind over terrains in our study case. Thank you, reviewer#1, for providing more information and references to complete the context of variational approach techniques on the wind retrievals. These statements were emphasized as follows: "For example, Collis et al. (2013) and Varble et al. (2014) use variational techniques to retrieve the winds via scanning Doppler radar. Also, the 3D variational techniques (3DVAR) for radar wind retrieval were developed by Shapiro and Potvin and are now available on the Python platform named PyDDA (Jackson et al. 2020). However, the terrains in their schemes were not significantly considered." in L80-84.

3. The quantitative evaluation of the squall line in earlier and later stages had included in the revision. The results reveal that the statistical consistency with only one time step was used in the previous analyses. Overall, scenario SCX also shows good performance, and the quantitative value of mean MB and RMSD are relatively smaller than other scenarios (cf. Fig. 12 and Table 4). The minor changes to the value of winds, the new figures and table (Figs. 11, 12, and Table 4), and the additional descriptions were revised in the manuscript as follows:
L260-261: The performance of WISSDOM wind retrieval was analyzed for this case study at 04:30, 05:30, and 06:30 UTC as the squall line moved from the ocean, coast to the land, respectively.
L535-537: The RWPs provided the average vertical profiles of U-winds, V-winds, and W-winds, allowing the WISSDOM winds to be compared above these three RWPs during three stages from 04:30 to 06:30 UTC on 2 August 2020.

[Figure]

Figure 11. (a) Average vertical profiles of the U-wind speed (thick black line) observed at RWP1 at 04:30, 05:30, and 06:30 UTC on 2 August 2020. The thin lines with numbers and colors indicate different scenarios. Number 1 colored black indicates scenario S (see Table 3). Numbers 2, 3, 4, 5, and 6 colored red, blue, green, pink, and orange indicate the scenarios C, X, SC, SX, and SCX, respectively. (b), (c) The same as (a) but for V-winds and W-winds. (d), (e) and (f) are the same as (a), (b), and (c) but for RWP2. Note that only two time steps (04:30 and 05:30 UTC) were included in (f). (g), (h) and (i) are the same as (a), (b), and (c) but for RWP3.

L555-558: …[missing and smaller]…, …[were ~5 m s$^{-1}$].

L561-562: Smaller differences were found above 6 km MSL of only ~0.5 m s$^{-1}$, note that RWP-2 W-winds were not included at 06:30 UTC due to data missing.

L566-568: ...[scenario X (~20 m s$^{-1}$)]...; ...[exceeding 6 m s$^{-1}$]...

L576-581: ...[WISSDOM scenario (thin black lines)]...; ...[around 1 m s$^{-1}$ and 3.5 m s$^{-1}$, respectively, between each scenario (red lines)]. [The MB for the horizontal wind speeds was ~3.5 m s$^{-1}$]...; [The MB values]...[RWP3 (less than 2 m s$^{-1}$)]...[of 1.6 m s$^{-1}$ in scenario S]...[than 3 m s$^{-1}$ for scenario X (blue lines)].

L581-584: Although the lowest mean MB of horizontal winds (i.e., counting U-winds and V-winds) is 0.93 m s$^{-1}$ for scenario C (thick black line in Fig. 12a), a little higher of mean MB (1.01 m s$^{-1}$) was observed between the observations and scenario SCX.

L585-588: However, the MB for the W-winds ranged between ~ 2.5 m s$^{-1}$ in the comparison between RWP1 and the WISSDOM scenarios (red line in Fig. 12b), and the lowest mean MB of W-winds is 1.1 m s$^{-1}$ for scenario SCX (thick black line in Fig. 12b).

[Figure]

Figure 12. (a) Mean bias (MB) of the U-wind speed (solid lines marked with U) and V-wind speed (dashed lines marked with V) for every scenario in WISSDOM and for the sounding (black lines marked with S), RWP1 (red lines marked with 1), RWP2 (green lines marked with 2), and RWP3 (blue lines marked with 3) data. The thick black line indicates the mean MB of U-winds and V-winds. (b) The same as (a) but for W-wind speed (solid lines marked with W) and mean MB of W-winds. (c) The same as (a) but for the root mean square difference (RMSD), but The thick black line indicates the mean RMSD of U-winds and V-winds. (d) The same as (c) but for the W-wind speed (solid lines marked with W).

L599: …[(thin black lines),]…; …[and ~2–4 m s$^{-1}$]…
L603-604: The lowest mean MB of horizontal winds is 1.57 m s$^{-1}$ for scenario SCX (thick black line in Fig. 12c).
L604-607: Fig. 12d presents the RMSD for the W-winds between RWP1 and RWP2. The RMSD was ~0.7 m s$^{-1}$ and ~2.5–3.0 m s$^{-1}$ at RWP2 and RWP1, respectively, in comparison with the WISSDOM scenarios. The lowest mean MB of W-winds is 1.5 m s$^{-1}$ for scenario SCX (thick black line in Fig. 12d).
L610: [Overall]…

L616:Table 4. Comparisons between the sounding and RWPs for each scenario during 04:30 and 06:30 UTC on 2 August 2020.

| | Mean Bias (MB, m s$^{-1}$) | | | Root Mean Square Difference (RMSD, m s$^{-1}$) | | |
|---|---|---|---|---|---|---|
| | U-winds | V-winds | W-winds | U-winds | V-winds | W-winds |
| S | 0.1 / 1.6* | 0.2 / 2.6 | — / 1.3 | 1.6 / 3.5 | 1.6 / 4.1 | — / 1.7 |
| C | 1.2 / 1.4 | 1.1 / 1.6 | — / 1.3 | 2.5 / 3.4 | 1.6 / 3.6 | — / 1.6 |
| X | 0.8 / 0.9 | 0.8 / 2.6 | — / 1.5 | 1.5 / 4.5 | 2.1 / 4.5 | — / 1.6 |
| SC | 0.6 / 1.2 | 0.7 / 2.1 | — / 1.2 | 1.7 / 3.2 | 1.7 / 4.0 | — / 1.7 |
| SX | 0.2 / 1.5 | 0.2 / 2.6 | — / 1.3 | 1.5 / 3.6 | 1.6 / 4.2 | — / 1.7 |
| SCX | 0.7 / 1.4 | 0.5 / 2.0 | — / 1.0 | 1.7 / 3.1 | 1.7 / 3.9 | — / 1.5 |

*Sounding / RWPs

**Major comments:**

Lines 87: Cha and Bell (2023) also added the IBM method to SAMURAI. Please mention their work in your literature review.

The descriptions of IBM in SAMURAI have been added in L88-89 as "Cha and Bell (2023) upgraded the SAMURAI by implementing IBM so that the wind can be better retrieved over complex terrain.". This article has been cited in the texts.

The authors should also mention the 3DVAR work done by Shapiro and Potvin that are now in PyDDA (Jackson et al. 2020). These works should also be mentioned in the literature review.

This article has been cited in L82-84, and we also remarked on this work as "Also, the 3D variational techniques (3DVAR) for radar wind retrieval were developed by Shapiro and Potvin and are now available on the Python platform named PyDDA (Jackson et al. 2020).".

**Minor/technical comments:**

Line 100-103: Run on sentence.

The sentences were rewritten for clarity. Please check them in L104-107: "Although S-band radar usually covers a wide area, radar data may be missing at lower levels far from the radar site. At the same time, the radar gate volumes become larger if the gate locations are too far from the radar site, leading to ambiguous radar observations, which is why the combination of radars was important.".

Line 104: "lower" should be "coarser"
Revised as suggestion.

174: "frozing" should be "freezing"
Revised as suggestion.

Figure 2: The station measurements are difficult to read on the figure. I would suggest removing some and making the font size bigger, or removing all of them.

The Korea Meteorological Administration (KMA) officially provided these figures. The station measurements cannot be modified to keep the original information from the KMA.

The authors tried to clarify the revised figure and emphasized the locations of the short front and the Korean Peninsula by red circles. Please find the revised figures below. Thanks.

In L236:

[Figure]

Figure 3. Korea Meteorological Administration surface analysis maps obtained at (a) 00:00 UTC and (b) 12:00 UTC on 2 August 2020. The purple shading indicates areas containing high moisture, while the arrows indicate the possible direction of movement. The red circle marked the locations of the Korean Peninsula and the short front.

Line 216: Extra "."
Revised as suggestion.

Line 361: "An"
The word was modified as "A".

Line 545: "leading edge"
The word has been added in the texts.

Figure 4: "reflectivity"
The word was modified.

---

## Author Comment (AC3)

**egusphere-2025-1908**
**Responses (highlighted in red) to Referee#2**
**25 August 2025**

The manuscript *Advantages of using multiple Doppler radars with different wavelengths for three dimensional wind retrieval* by Tsai et al. presents a valuable effort to evaluate the use of different radar bands configuration in the WISSDOM multi-Doppler configurations for observing a severe weather event around Seoul. However, several aspects of the study require clarification to enhance its scientific rigor and broader applicability. In particular, the intent of the analysis, and the methods used to evaluate the results. The following major and minor comments aim to help the authors improve the clarity, accuracy, and overall impact of their work.

We appreciate Referee#2 providing helpful and insightful comments in this round, which help us to improve the manuscript substantially. We have carefully checked your comments. The authors have emphasized the indentation in the revision. In addition, more discussions on the scan strategies, terrain effects, and the setting up in WISSDOM have been revised in the revision. A set of responses to your comments is provided below. Specific locations of modified portions (marked as underlines) were also noted as the number of lines in the revised manuscript.

**Major comments:**

1. There is some ambiguity regarding the intent of the study: is this a case study analysis or a broader investigation into network design? If this is intended as a single-case study, could the authors clarify why conclusions about radar network configuration are generalized? For example, the statement in L550–551 seems to imply a broader applicability, but the findings are drawn from a single squall-line event occurring in a specific region. Would it be more accurate to frame these conclusions in the context of this particular setup?

   Thanks for Referee#2 pointing out this important point. Yes, this is a case study, and the retrievals should vary case by case. However, the authors intend to evaluate the performance of WISSDOM retrievals by choosing a significantly severe weather phenomenon, and the main reasons are explained below and clarified in the revision.

First, the squall line is one of the severe weather systems. According to Tsai et al. (2023), the WISSDOM retrievals could be better evaluated from strong wind cases to examine potentially maximum errors. Therefore, a squall line case was chosen for further evaluations in this study. This statement has been emphasized in L152-154: "A squall line case was chosen for the evaluations because significant precipitation and strong winds may help us to examine the potential errors in the retrieval winds (Tsai et al. 2023).".

Second, the authors intend to ensure that the WISSDOM retrieval of precipitation and wind structures of the squall line case are corrected. Then, the quantitative evaluations could be performed. How can you make sure the structures of a squall line are right? The typical structure of a squall line (Houze et al., 1989) can be utilized for the comparison. The authors have emphasized these descriptions in the revision as follows. L263-266: "First, this study had qualitatively checked the characteristics of precipitation and wind patterns (i.e., return flow, etc.) before quantitatively evaluating the accuracy of the retrieved winds. This step can initially confirm the reliability of retrievals in WISSDOM.".

Third, once the potential errors can be estimated in a significant case, these procedures and results in this study can be easily expanded and serve as a reference for more similar cases, such as mesoscale convective systems and others, like typhoons or afternoon thunderstorms. These intentions were also emphasized in the revision; please find it in L702-704: "Although this is a case study, the performance of WISSDOM retrievals may vary case by case. In the future, other weather systems such as typhoons and fronts can be included in the analysis.".

2.  Section 2.4 could benefit from clarification. The content in lines 282–291 might be presented more clearly, to enhance conciseness, and improve readability. Additionally, lines 292–296 repeat information already discussed in the previous paragraph.

Thanks for Referee#2's comments. The authors have rewritten the descriptions clarity, Please find it in L344-355: "Because the sounding site (#47199) and three radar wind profilers (RWP-1–3) were collocated in the WISSDOM domain. Thus, the mean bias (MB) and root mean square deviation (RMSD) between retrieved WISSDOM winds, sounding, and RWP-1–3 observations were selected as evaluation metrics in the present study, as Tsai et al. (2023) evaluated the wind retrievals in WISSDOM. Since the vertical spatial resolution of the sounding

observations was about 3~5 m, associated with the rate of rise of the sensors (3~4 m s$^{-1}$), the data had to interpolate to 250 m for fitting the vertical grid spacing of WISSDOM. The MB and RMSD were estimated by tracking the exact rising path of the sounding sensor, because the sounding tracks are not usually right on the grid point of WISSDOM. Therefore, the sounding observations near the closest grid point in WISSDOM and their retrieval winds were selected to estimate the MB and RMSD. The sounding launching time at 06:00 UTC on 2 August 2020 was selected for further evaluations (i.e., the closest time to the WISSDOM analysis period from at 05:30 UTC).".

The redundances words in L292-296 have been removed, the revised descriptions can be found in L356-358: "The RWPs were fixed stations that provide vertical 3D wind information from the surface. The RWP observations were interpolated to 250 m to allow for comparison with the WISSDOM derived winds during the same time steps at 04:30, 05:30, and 06:30 UTC.".

3. Understanding how the data are gridded and whether filters are applied is essential for assessing quality. For instance, are filters in WISDOMM applied that might mitigate aliasing errors? Additionally, how are the gridding parameters (e.g., horizontal/vertical resolution, interpolation scheme) chosen, and how might these influence the retrieved wind field?

This is a valid point, it is important to explain more about the quality control and treatments for the radar data before implementing it in WISSDOM. Three procedures were needed to apply, removing non-meteorological and useless signals, the lowest radar data, and unfold radial winds. Therefore, A fuzzy logic QC algorithm was employed, the topographic height was considered, and the winds were unfolded. In this stage, the radar data remained the same grid size as the original size for further interpolation to WISSDOM. The variational scheme for the wind retrievals requires high-quality inputs; these procedures may have deduced the potential error rising in WISSDOM as much as possible. The descriptions have been explained in the revision as follows. L186-193: "In particular, the lowest radar data (radar reflectivity and radial winds) was obtained using a relatively high discrimination threshold. The radar data was eliminated if there was an eclipse of topography of more than 10% to retain only reliable data. The radial winds were unfolded if the radial wind was folded (i.e., over the Nyquist velocity for each radar). In this stage, the QC radar data remains the same grid size as the original coordinate. Then the useful and reliable radar data can be confidently

interpolated to mitigate the possible errors mainly produced by non-meteorological and useless signals in further WISSDOM retrievals (the details in Sec 2.3).".

4. **"**The bow shape of the squall line is not particularly evident across the different scenarios, and the assertion that SCX provides the most "accurate" representation of the wind" (L 353-355) field appears speculative, especially since it's based on visual inspection. Could the authors clarify the basis for determining "accuracy" in this context?

The bow shape in convection usually accompanies stratiform precipitation behind the leading edge, as it may be linked to the rear-to-front flow enhanced by evaporation cooling. Except for the squall line in mesoscale convective systems, bow shape can be found in squall line-like tropical cyclone rainbands, and they were both observed rear-to-front behind the leading edge (Yu and Tsai, 2013; Yu et al., 2020; Swastiko et al., 2024). In our case, that is our point in discussing more reasonable strictures of the studied squall line. However, the authors modified the word "accurate" to become "reasonable" in the texts to clarify our intention. Those descriptions have been revised in L420-424: "These horizontal airflow and precipitation structures closely matched the typical characteristics of the squall line in mesoscale convective systems (Swastiko et al., 2024) and squall line-like bow echo in tropical cyclone rainband (Yu and Tsai, 2013; Yu et al., 2020), meaning that the scenario SCX may have produced the reasonable and representative wind field synthesis.".

5. The phrase "highlighting the positive impact of adding C and X-band radar obs to S-band radars" (L433) suggests a generalized improvement, but this is essentially a typical gap-filling outcome.

This is a valid point, thank you. Initially, the authors tried to explain the advantage of short wavelength radars in this case; however, this phrase is unclear. In this case, the C- and X-band radars were located at lower elevations and thus they can provide good coverage at lower levels. However, this case is not a general condition due to varying locations and elevations for each short wavelength radar with different squall line cases. The authors revised the phrase for charity in L503-505: "These results had characteristics similar to a typical squall line in this case,

thus highlighting the positive impact of adding C- and X-band radar observations to S-band radars as they can provide sufficient data at lower levels.".

6. Maybe it would be good to clarify that while this set-up is beneficial in this specific case, similar improvements may not occur for other cases? (e.g., to acknowledge that this may vary case-by-case, rather than presenting it as a general characteristic?)

Thank you for the suggestion; it will make our intention more straightforward. The authors have emphasized this statement in the revision. Please find them as follows:

L503-505: "These results had characteristics similar to a typical squall line in this case, thus highlighting the positive impact of adding C- and X-band radar observations to S-band radars as they can provide sufficient data at lower levels.".

L699-672 (Sec. 3.4 Discussions): "Thus, the C- and X-band radars are essential in WISSDOM synthesis for more accurate 3D wind retrieval if they can cover more lower-level areas. Based on the setup, it is beneficial in this case study, however, the performance of WISSDOM retrievals will need more evaluations for the other cases and weather phenomena.".

L702-704 (Sec. 4 Conclusion): "Although this is a case study, the performance of WISSDOM retrievals may vary case by case. In the future, other weather systems such as typhoons and fronts can be included in the analysis.".

7. The statement about C-band radars providing more near-surface data needs clarification (L397). This behavior is not general and is highly dependent on radar configuration and terrain. Could you be more specific about the setup used in this study (e.g., scanning strategies, beam elevation angles, and terrain impact)? How does WISDOMM deal with terrain?

The coverage and topographic blocked areas of each radar at different levels have been analyzed in a new figure 2. The C-band radar can provided better coverage at 0.5 km MSL rather than S-band radars. Thus, the statement can been verified, then the sentence was revised for clarity in L466-467: "However, the C-band radars produced more radar observations near the surface (cf. Fig. 2)".

Since this behavior may have the differences case-by-case, the scanning strategies, beam elevation angles, and terrain impact were added and discussed in the revision. Please find the revised parts in:

L168-174(canning strategies, beam elevation angles): "The temporal resolution for each radar volume scan was 10 min except for CIIA (XDJK, XMIL, and XSRI), which was around 6-7 min (~15 min), the complete volume scan can be synchronized every 30 min from the selected radars. In a complete volume scan of each radar, the PPI (plan position indicator) elevation angles were concentrated between −0.4° and 20° (45° for CIIA), the details of the elevation angles can be found in Table 1. Fundamentally, the radars used in this study are mostly synchronized in similar scanning strategies, even though they were operated from different departments of governments and universities.".

L183 (Table 1):

Table 1. Specifications for the radars used in the present study

| | Longitude (°E) | Latitude (°N) | Radar Height (m) | Wave length (cm) | Beam Width (°) | Nyquist Velocity (m s⁻¹) | Range Resolution (m) | Max Range (km) | Volume scan Interval (min) | Elevations (°) |
|---|---|---|---|---|---|---|---|---|---|---|
| SGDK | 127.43 | 38.11 | 1066 | 10 | 0.89 | 64.3 | 250 | 250 | 10 | −0.4 0.0 0.3 0.8 1.4 2.5 4.2 7.1 15 |
| SKWK | 126.96 | 37.44 | 615 | 10 | 0.93 | 68.3 | 250 | 250 | 10 | −0.2 0.0 0.3 0.8 1.5 2.6 4.4 7.3 15 |
| SBRI | 124.62 | 37.96 | 170 | 10 | 0.96 | 64.7 | 250 | 250 | 10 | 0.1 0.4 0.8 1.4 2.2 3.4 5.1 7.6 15 |
| SKSN | 126.78 | 36.01 | 212 | 10 | 0.90 | 67.9 | 250 | 250 | 10 | 0.0 0.3 0.7 1.3 2.1 3.2 5.0 7.6 15 |
| CIIA | 126.36 | 37.46 | 142 | 5 | 0.53 | 29.7 | 250 | 130 | ~6 | 0.7 1.0 1.5 2.1 2.9 4.0 6.0 8.0 11 15 21 28 36 45 |
| CSAN | 126.49 | 36.70 | 45 | 5 | 0.95 | 47.9 | 250 | 130 | 10 | 0.5 0.9 1.4 2.0 2.6 3.4 4.5 5.9 7.6 10 13 20 |
| XKOU | 127.02 | 37.58 | 136 | 3 | 0.53 | 18.0 | 60 | 40 | 10 | 3.0 3.6 4.3 5.1 6.1 7.2 8.6 10.2 12.2 14.4 17 20 |
| XYOU | 126.93 | 37.56 | 79 | 3 | 0.45 | 18.0 | 60 | 40 | 10 | 2.5 3.0 3.7 4.4 5.4 6.5 7.8 9.4 11.4 13.6 16.4 20 |
| XDJK | 126.09 | 37.25 | 116 | 3 | 1.26 | 44.8 | 150 | 75 | 15 | 1.5 2.1 3.0 4.2 5.8 7.9 15 |
| XMIL | 126.44 | 36.93 | 295 | 3 | 1.26 | 44.8 | 150 | 75 | 15 | 0.8 1.4 2.2 3.5 5.2 7.9 15 |
| XSRI | 126.90 | 37.35 | 435 | 3 | 1.26 | 44.8 | 150 | 75 | 15 | 0.8 1.4 2.2 3.4 5.2 7.9 15 |

L193-203(terrain impacts): "Figure 2 shows the radar coverage and topographic blockage at constant high levels. The mountains are not sufficiently high in South Korea; therefore, there were no significant terrain blockages in the WISSDOM domain (Figs. 2a and 2b). In addition, the S-band radars cannot provide sufficient observations at lower levels because they are usually located at higher elevations and far from the WISSDOM domain. Although the C-, X-band radar observations were also limited at the lowest level, they can provide good coverage from 0.5 to 1 km MSL (Figs. 2b and 2c). The overlay area of radars was increased from 2 or 3 radars to 5 or 6 radars in the WISSDOM domain below 1 km MSL (contributed mainly by short wavelength radars), then the overlay area was expanded and occupied most areas with 5~7 radar numbers in WISSDOM domain from 1 km, 5 km up to 10 km MSL (Figs. 2d-2f).".

L204 (Figure 2):

[Figure]

Figure 2. (a) The coverage area and topographic blockage of the radar observations were explored at 0.25 km MSL (Mean Sea Level) height, the color shading indicates the overlay areas counting by the radar numbers. The location of S-, C-, and X-band radars were marked by dark blue, light blue, and green triangles, respectively. The black box is the WISSDOM domain as same as in Fig.1a. (b), (c), (d), (e), and (f) are the same as (a), but for the height at 0.5, 1, 2, 5, and 10 km MSL.

In this study, the IBM (immersed boundary method ) was adopted in WISSDOM, this algorithm allows for the extraction of closer information near the surface for each grid in WISSDOM. The details were emphasized in L321-330: "This algorithm allows for the extraction of closer information near the surface for each grid in WISSDOM. As it is known that observations are often lacking near the surface, it may be limited to computing and simulating atmospheric variables at the lower boundary, especially over terrains. Therefore, WISSDOM kept and computed the winds from the lowest grid by adopting the IBM; the results of the retrieved winds can better reflect the real situations at the lower boundary over complex terrain up to higher levels. Those advantages are the reason why SAMURAI has been upgraded by applying the IBM (Bell et al., 2012; Cha et al., 2023), and MUSCAT (Chong and Bousquet, 2001) has also applied the IBM, even for further study on tropical cyclone (Cheng et al., 2025).".

8. A reference/discussion to lower-boundary limitations due to topography and data availability, and how those may impact wind retrievals would help contextualize the limitations of the analysis.

The observations are usually lacking data at the lower boundary, especially on the surface over complex terrains. The IBM algorithm can help extract more information near the surface and is applied in the variational-based algorithm for wind retrievals like WISSDOM, SAMURAI, and MUSCAT. The authors have discussed and explained these points of view. Several references were also provided in the revision in L319-330: "The other advantage of WISSDOM is that it applies IBM for computing the winds over complex terrain (Liou et al., 2012). IBM can simulate the fluid patterns over a complex geometry on Cartesian coordinates (Peskin, 1972). This algorithm allows for the extraction of closer information near the surface for each grid in WISSDOM. As it is known that observations are often lacking near the surface, it may be limited to computing and simulating atmospheric variables at the lower boundary, especially over terrains. Therefore, WISSDOM kept and computed the winds from the lowest grid by adopting the IBM; the results of the retrieved winds can better reflect the real situations at the lower

boundary over complex terrain up to higher levels. Those advantages are the reason why SAMURAI has been upgraded by applying the IBM (Bell et al., 2012; Cha et al., 2023), and MUSCAT (Chong and Bousquet, 2001) has also applied the IBM, even for further study on tropical cyclone (Cheng et al., 2025).".

9. Given the limitations of using the sounding as "ground truth" (only valid for that grid point), might it be more robust to compare the dual-Doppler retrievals with high-resolution model output? Could a sensitivity analysis be conducted, testing the effects of vertical coverage and radar configuration on retrieval quality? This could provide a more systematic understanding of the strengths and limitations of the setup, especially in the absence of in-situ storm-scale observations.

The sensitivity test was performed in scenario SCX using sounding and reanalysis datasets (named LDAPS, it's a regional model of Korea Meteorological Administration, the details can be found in Sec. 3.2.4 of Tsai et al., 2023). The results in the discrepancies of the retrieved winds are shown in the figure below (Figure RC2.1), the U-, V-winds had relatively small differences just in $\pm$ 1 m s$^{-1}$ (Figs RC2.1a, b, d, e), and the U-, V-winds had relatively small differences just in around $\pm$ 0.2 m s$^{-1}$ (Figs RC2.1c, f).

[Figure]

Figure RC2.1 (a) The difference in the U-winds of scenarios SCX at 2 km MSL (Mean Sea Level) by implementing in-suit observations and reanalysis LDAPS datasets in background of WISSDOM. (b), (c) are the same as (a), but for V-, and W-winds, respectively. (d), (e), and (f) are the same as (a), (b), but for the average cross-section corresponding to the box along A-A'.

The descriptions about the compatibility of WISSDOM have been added in L310-312: "The discrepancies of retrieved winds were minor while the reanalysis datasets were applied in WISSDOM (not shown), and the results reveal compatibility in case of lacking in-situ storm-scale observations.".

10. L545-547 statement implies a core assumption, but it's actually fundamental to the validity of the analysis. Could the authors be more explicit about how the radar sampling strategy, scanning configuration, and network geometry impact the results? For example, how much blockage is present per radar? Are there areas that are not well sampled at low levels by the S-band radar, but are captured by the X-band system?

The radar sampling strategy, scanning configuration, and network geometry are key factors affecting the results significantly. The radars operated by different government and university departments were used in this study. Thus, the mentioned key factors were almost fixed due to their purpose in monitoring the weather, precipitation, and water resources.

The coverage and topographic blockage of radars for each level were computed to verify the potential influences in WISSDOM synthesis. The results indicate that short wavelength can provide better coverage at a lower level than S-band radar observations in this case. Therefore, this statement could be evidenced through this figure and related information (please check the details in our responses to your Major comments 7, thank you).

11. Understanding how the data are gridded and whether filters are applied is essential for assessing quality. For instance, are filters in WISDOMM applied that might mitigate aliasing errors? Additionally, how are the gridding parameters (e.g., horizontal/vertical resolution, interpolation scheme) chosen, and how might these influence the retrieved wind field?

Please referred our responses to your Majoy comments 3, thank you.

**Minor edits:**

L58-59: Please, clarify "measure radar reflectivity of the documentation of precipitation structures".

The sentence has been revised for clarity. Please find it in L56-58: "In particular, meteorological radars are widely used to measure radar reflectivity and radial velocity for determining precipitation structures and kinematic information of the weather systems."

L80-81: I suggest rewriting this line as it sounds like the variational method is uniquely a type of multi-doppler technique.

The sentence has been modified as a suggestion, please find the new on as "In particular, starting in the 2010s, mathematically variational approach techniques were utilized to retrieve winds gradually." in L79-80.

L84: Cha and Bell 2023 developed SAMURAI over complex terrain. This work should be mentioned here.

This article has been cited in the texts, and the descriptions can be find in L88-89: "Cha and Bell (2023) upgraded the SAMURAI by implementing IBM so that the wind can be better retrieved over complex terrain."

L113: Although the facts in the=is statement are correct, it reads as "smaller precipitation particle" detection and "gap filling" are directly correlated, when they are not. Please, rewrite this sentence. =

Thank for pointing the problems, the descriptions have been rewritten clarity as "In contrast, C- and X-band radars are less expensive and more mobile and sensitive to smaller precipitation particles. The shorter wavelength radars are ideal for gap-filling applications and provided more information even in light rain events." in L116-118.

L141: Please clarify this sentence, I don't understand the relationship between having more radars available and increasing the availability of thermodynamic fields being related.

Since the retrieved winds of WISSDOM can be used to derive thermodynamic winds by governing simple momentum and thermodynamic equations (Liou et al., 2019; Liou and

Teng, 2023), the result of thermodynamic fields is possibly linked to WISSDOM and radar observations. The descriptions have been explained by adding the sentences in L144-146: "Recently, Liou et al. (2019) and Liou and Teng (2023) derived thermodynamic fields using the retrieved winds of WISSDOM. Thus, the accuracy of derived results is linked to the data quality of radar observations."

The reference was also cited here and in the reference list.

L143: Please, change to read "storm dynamics and phenomena".
Revised as a suggestion.

L144: Please, clarify "their advantages".

The main advantages of using multiple wavelength radars are that they provide better coverage and high spatial resolution observations. The sentences have been rewritten for clarity, please find them in L149-152: "To address this gap, this study conducts a quantitative and systematic assessment of the advantages of using multiple wavelength radars, such as their ability to provide more coverage (especially at lower levels) and high spatial resolution observations. It allows us to evaluate the uncertainty, and accuracy of wind retrieval using independent wind observations.".

L151: There are two "area" words in the same sentence.
The word was replaced by "region".

L166: Please, change to read "spatial" instead of "horizontal".
The word has been revised.

L155: Change to read: automatic weather stations (AWS)
The word has been modified.

L168: Remove the small "s" after AWS.
The letter has been removed.

L174: Please, change to read "freezing".

The word has been corrected.

L193: Please, change to read "may be affected".

The word has been added.

L196: Please, change to read "convective" instead of "convection".

The word has been replaced.

L202: Please, change to read "The evolution of"

The word has been changed.

L207: Please, change to read ""stratiform precipitation was located behind the convective region."

These two words were revised.

L208: I believe this would benefit from a reference (after squall line).

Two important references of squall line studies (Houze, 1977; Houze et al., 1989) have been added in the texts.

L213: Maybe "stratiform precipitation areas"? Instead of "formations"?

The word has been replaced.

L218: Please, change to read "moved easterly".

The word has been changed.

L231: Please, change to read: Liou and Chang (2019)

Thanks for pointing the problem, this citation should be Liou and Chang (2009). The citation has been revised in the texts.

L263-264: Consider re-writing it. It is confusing as it is.

The sentence has been rewritten for clarity as "The AWS observations were bilinearly interpolated to the lowest grid point above the ground, and the horizontal distance weighted using a Gaussian distribution between the AWSs and each grid point." in L306-308.

Table 2: Under "Data Implementation- Background", change "linier interpolation" to "linear interpolation"
The word has been corrected.

L277: Might be useful to cite those studies.

Those studies (Liou and Chang, 2009; Liou et al., 2012, 2014, 2016, 2019, 2024; Tsai et al., 2018) have been cited in the manuscript.

L285: Change to read:" [...] domain, the mean bias (MB) [...]"
Revised as a suggestion.

L286: Please, clarify "[...] associated with the rate of rise of the sensors [...]". I assume it is related to the vertical velocity of the sounding sensor, but I don't understand the relationship between that and the sensor gridspacing (horizontally?) and the mean interpolation to 250m.

The word "grid spacing" may be confused here; it should be corrected to "vertical spatial resolution" for clarity. The original vertical spatial resolution of sounding data is related to the rising rate of the sounding sensor (about 3~5 m s$^{-1}$, with sampling frequency of 1 second). However, the vertical grid spacing was set as 250 m in WISSDOM; the sounding data had to be interpolated from 3~5 m to 250 m for running WISSDOM. The description has been revised in L348-350: " Since the vertical spatial resolution of the sounding data was about 3~5 m, associated with the rate of rise of the sensors, the data were interpolated to 250 m for fitting the vertical grid spacing of WISSDOM."

L287-288. The sentence does not have much sense on its own. No verb is found.
The verb has been added in the sentence, thanks for finding the problem.

L300: Move the RWP explanation to L292, when it is referred to. It would be easier for the reader to understand.

The RWP explanation was move to the begging of this paragraph, thanks for the suggestion.

L314: I suggest referring the gust front position to the main storm (e.g., 50 km away on the leading edge of the main squall line, at X=125km).

Thank you for this good suggestion! The descriptions have been added in L376-379.

L340: Please, change to read "reflectivity", not "relativity". Same in L403 (Figure 7 caption).

Thank you, the word has been modified.

L348: There seems to be an inconsistency regarding the convergence area: if the scenario with only S and C-band data does not exhibit this feature, what, specifically, is influencing it in the other configurations?

The main reason is that scenario SC provided sufficient radar observations in WISSDOM synthesis; the explanations can be found in the figure and descriptions below. Figure RC2.2a shows that scenario S had poor coverage of radar reflectivity at 500 m MSL except for the smaller northwestern corner of the study domain. Scenario C provided better coverage of radar reflectivity at 500 m MSL except for the northwestern corner of the study domain (Figure RC2.2b). The convergence area (i.e., X = ~0–50 km, Y = ~130 km in Figure 5d) can be depicted when combining S-and C-band radar observations at lower levels (Figure RC2.2c). The convergence area cannot be constructed by considering individual contributions from the S-and C-band radars (Figures RC2.2d and RC2.2e); however, relatively stronger downdraft and updraft were found coincident with the convergence area in Scenario SC (Figure RC2.2f). The results indicated the importance of multiple wavelength radars in WISSDOM synthesis. These statements were emphasized in the manuscript in L413-415 as "Although these signatures were not observed in scenarios S and C, the convergence area can be produced due to better coverage of C-

[Figure]

band radar combining part of the S-band radar observations at lower levels (not shown).".

Figure RC2.2. Retrieved radar reflectivity (color shading, dBZ), and storm-relative flow (vectors) at 0.5 km mean sea level (MSL) obtained from WISSDOM for scenarios (a) S, (b) C, (c) X, (d) SC, (e) SX, and (f) SCX.

L361: Please, change to read "A less".
The word had been changed.

L388: I don't understand the reference to this threshold in this sentence. Please, clarify.

The thresholds of 25~40 dBZ are usually adopted to identify the intense precipitation areas, depending on the cases. In this case study, the threshold of 25 dBZ was selected because the boundary of intense precipitation and flow structure can be identified for each scenario. In addition, the interval of the color bar was modified by every 5 dBZ to check the squall line characteristics easily. The descriptions were emphasized in the texts in L456-458: "Precipitation and flow structures from scenario S (Fig. 8a) closely resembled those of a typical squall line (Fig. 7), using a radar echo threshold of 25 dBZ, as the intense precipitation and significant flow structures can be successfully identified in this case.".

L415: Please, remove "are".

The word has been removed.

L449: Please, remove the second "changes".

The word has been removed.

L545: Please, change to read "leading edge".

The word has been revised.

L563: It would be beneficial to indicate that the colorbar is scaled differently (or scale them all to the same range).

Thank you for the suggestion. The color bar was modified in the figures (i.e., Figures 13 and 14) to make the differences easier to see with the eye. The revised figures can be found below.

[Figure]

Figure 13. (a) The difference in the radar reflectivity between scenarios SCX and S (S is subtracted from SCX) at 2 km MSL. (b), (c) and (d) are the same as (a), but for U-, V-, and W-winds, respectively.

[Figure]

Figure 14. Same as Fig. 13 but for the average cross-section corresponding to the box along A-A' in Fig. 13.

- Reference:

Cha, T., and M. M. Bell, 2023: Three-Dimensional Variational Multi-Doppler Wind Retrieval over Complex Terrain. *J. Atmos. Oceanic Technol.*, **40**, 1381–1405, https://doi.org/10.1175/JTECH-D-23-0019.1.

The reference has been added in the list, thank you!

---

## Author Response (AR2)

**egusphere-2025-1908 Responses (highlighted in red) to Referee#1 11 October 2025**

I would still recommend a more objective approach to counting the updraft cores in their analysis, as just saying "it's more obvious from the figure" is not scientifically robust.

Thanks for Referee#1's comments. Since the performance of retraveled vertical velocity is still ongoing works for getting accurate one, qualitative definition was suitable for describe the vertical structure in present study. Therefore, the author have tied to provide a specific and reasonable definition for the updraft cores in the revision. The modified definition of updraft cores is that the vertical velocity is larger than 1.5 m s-1 with upward extension at least 5 km tall in the squall line.

This definition has been added in the texts as:

L498-499: Only one updraft core (defined as a vertical velocity over 1.5 m s-1 with upward extension at least 5 km tall, marked in dark orange color) was presented in scenario S (Fig. 9a),.....

**egusphere-2025-1908 Responses (highlighted in red) to Referee#2 11 October 2025**

Dear authors,

I have added some minor revisions that refer to the items listed in the first review under "Major Comments".

Thanks for Referee#2's comments. Please find our responses and the revised part as mark as red below.

1. Thank you for adding the clarifying points. I still have a comment regarding the concluding remarks. I am assuming that not only the performance of WISSDOM but the configuration would need to be changed for other systems, is that correct? If so, I would also add this in the concluding remarks (e.g., after your comment in L702-704). This would probably clarify that this finding is a great step forward but has only been tested in a squall

line-type system, geographically positioned so the current network and WISSDOM configuration has a positive result, but that for other cases, that configuration might change. Performance maybe does not change under another configuration, correct? It would only be under this configuration.

That should be stated clearly.

Also, in L702-704, did you mean to say, "Although the results are positive in this case study"?

Thanks for Referee#2 pointing out this problem. This is a valid point. Follow the reviewer comments, the authors provided the statement that the configuration may change with different cases or weather systems. The revised descriptions have been added in the revision as:

L706-709: Although the results are positive in this case study, the configuration of WISSDOM retrievals may vary case by case. This finding is a great step forward but has only been tested in a squall line-type system, geographically positioned so the current network and WISSDOM configuration has a positive result, but that for other cases, that configuration might change.

2. I suggest a review of the final text to ensure that there are no grammatical and spelling errors. For instance, the first sentence, does not make sense at is it right now. It is unfinished: "Because the sounding site (#47199) and three radar wind profilers (RWP-1–3) were collocated in the WISSDOM domain."

The sentence have been revised completely as:

L347-351: Because the sounding site (#47199) and three radar wind profilers (RWP-1–3) were collocated in the WISSDOM domain, the mean bias (MB) and root mean square deviation (RMSD) between the retrieved WISSDOM winds, soundings, and RWP-1–3 observations were selected as the evaluation metrics in the present study, following the same approach used by Tsai et al. (2023) in evaluating wind retrieval in WISSDOM.

Also, please, change to read: "[...] the data had to be interpolated [...] "
Revised as comment in L353.

- 3. Please, clarify:
- Lowest radar data. Do you mean lowest elevation?
- I would suggest changing "eclipse of topography" by "topography blockage".

Short expansions and the phrase have been revised as suggestion in L190-192: The lowest available radar data (i.e., useful radar reflectivity and radial velocity above ground) were obtained using a relatively high discrimination threshold. Radar data for each there was a topography blockage of more than 10% were removed to retain only realistic data.

- \_• Please, change "radial winds" by "radial velocity" anytime in the text that is related to the direct radar moment. We do not obtain winds directly from radars. The word has been revised throughout the manuscript.
- Whas the unfolding method and the blockage removal manual? Or did you use any specific algorithm?

Yes, the algorithm was developed by Cho et al., 2006; Ye et al., 2015. They provided a standard QC procedure in Korea radar observations. The articles gave been cited in the text, and the reader can get the information easily from their studies.

4. Would "realistic" fit better? (e.g., close to reality). The word has been replaced.

Please, make sure also to have a final English grammar review of the entire document for consistency and before final submission. Some sentences could use some rewriting, and others are lacking verbs or consistency.

The English grammar in the final version have been edited through the native English speaker. The issue of lacking verbs should be disappeared in the revision.

Thank you.